# Optical control of PIEZO1 channels

Francisco Andrés Peralta [1,2], Mélaine Balcon[1], Adeline Martz[1], Deniza Biljali[1], Federico Cevoli[1], Benoit Arnould[1], Antoine Taly[3,4], Thierry Chataigneau[1] & Thomas Grutter [1,2] ✉

PIEZO proteins are unusually large, mechanically-activated trimeric ion channels. The central pore features structural similarities with the pore of other trimeric ion channels, including purinergic P2X receptors, for which optical control of channel gating has been previously achieved with photoswitchable azobenzenes. Extension of these chemical optogenetics methods to mechanically-activated ion channels would provide tools for specific manipulation of pore activity alternative to non-specific mechanical stimulations. Here we report a light-gated mouse PIEZO1 channel, in which an azobenzene-based photoswitch covalently tethered to an engineered cysteine, Y2464C, localized at the extracellular apex of the transmembrane helix 38, rapidly triggers channel gating upon 365-nm-light irradiation. We provide evidence that this light-gated channel recapitulates mechanically-activated PIEZO1 functional properties, and show that light-induced molecular motions are similar to those evoked mechanically. These results push the limits of azobenzene-based methods to unusually large ion channels and provide a simple stimulation means to specifically interrogate PIEZO1 function.

In eukaryotic cells, PIEZO1 and PIEZO2 rapidly convert applied force into electrochemical signals[1,2], a process that underlies important physiological functions, such as touch sensation[3], cell volume regulation[4], proprioception[5], and baroreception[6]. PIEZO channels are the largest plasma membrane ion channel complexes identified thus far[1]. Totaling over 7500 amino acids, they assemble into trimers and form a central pore permeable to cations, including calcium ($Ca^{2+}$). Each subunit contains 38 predicted transmembrane (TM) helices. Upon application of pressure, PIEZO1 rapidly permeates cations, within a few ms, through an open channel conformation with a single-channel conductance of ~30 pS (refs. [2,7]). This transient open state is followed by an inactivated state, which takes place in the presence of continued stimulation[1,8]. Besides mechanical activation, it has been shown that synthetic chemical compounds, such as Yoda1, Jedi1, and Jedi2 also activate PIEZO1 channel, but much more slowly than mechanical stimulation (several seconds)[9–11]. It is thus unknown whether other stimuli, such as light, can be used to activate PIEZO channels. The development of alternative stimulation methods based on new molecular methods would be instrumental in endowing PIEZO with the ability to respond to external stimuli other than non-specific and technically demanding mechanical stimulations.

Recent high-resolution structures of both PIEZO1 and PIEZO2 reveal a three-blade propeller architecture, with a central ion pore[12–17]. The three blades, thought to sense mechanical forces, are formed by nine repetitive elements of four-helix bundles (only six are resolved in PIEZO1 structure) and extend outward within the lipid bilayer by spiraling away from the center of the pore, giving an unusual dome-shaped architecture[13] (Fig. 1a). The central pore is lined by the final two C-terminal TM helices, termed the outer TM37 and inner TM38 helices, that are connected by an extracellular cap domain. This topology is reminiscent of that of other trimeric ion channels, including acid-sensing ion channels (ASIC) and P2X receptors (P2X), in which the two pore-lining TM1 and TM2 helices (equivalent to TM37 and TM38, respectively) are also connected by an extracellular ectodomain. Structural comparison between these channels reveals a similar trimeric pore arrangement of inner and outer helices, despite their

[1]Équipe de Chimie et Neurobiologie Moléculaire, Laboratoire de Conception et Application de Molécules Bioactives (CAMB) UMR 7199, Université de Strasbourg, Centre National de la Recherche Scientifique, Faculté de Pharmacie, 67401 Illkirch, France. [2]University of Strasbourg Institute for Advanced Studies (USIAS), 67000 Strasbourg, France. [3]Laboratoire de Biochimie Théorique, CNRS, Université Paris Cité, UPR 9080 Paris, France. [4]Institut de Biologie Physico-chimique, Fondation Edmond de Rothschild, Paris, France. ✉e-mail: grutter@unistra.fr

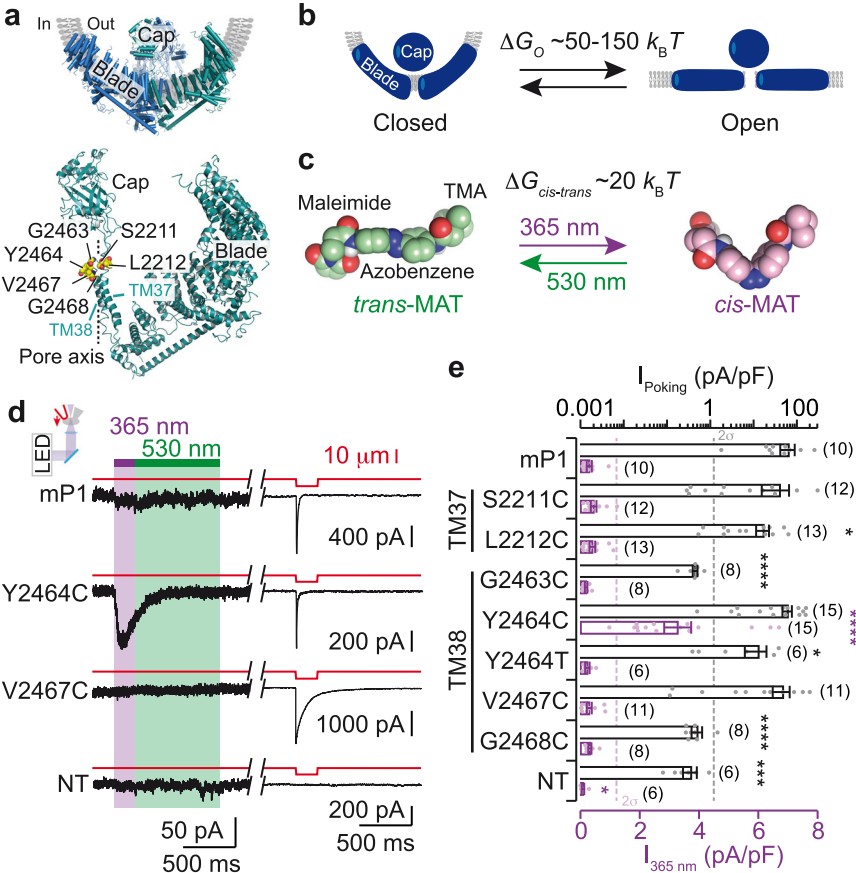

**Fig. 1 | Engineering light-gated PIEZO1 channels. a–c** Design of photochemical activation of mP1. **a** Side view of a cartoon model of mP1 structure (PDB ID: 5Z10)[14] showing the trimer with featured domains (top) and one of the three subunits with selected residues for cysteine mutation in TM37 and TM38 (bottom). The approximate position of the membrane and pore axis is indicated. **b, c** Range similarity (within an order of magnitude) between $\Delta G_O$ and $\Delta G_{cis\text{-}trans}$. $\Delta G_O$ is the energy difference between a flat, presumably open conformation, and a curved, presumably closed conformation of mP1, in the absence of mechanical force[22]. $\Delta G_{cis\text{-}trans}$ is the energy difference between *cis* and *trans* configurations of the azobenzene[24]. Molecular models (displayed as spheres) of *trans*-MAT and *cis*-MAT showing different moieties of the photoswitch (TMA, trimethylammonium). Carbon atoms are color-coded according to irradiation wavelengths. **d** Representative traces of inward currents in the whole-cell configuration in response to light or poking stimulations of MAT-treated HEK-P1KO cells expressing the indicated cysteine mutants or mP1. Each cell, held at a holding potential of −80 mV, was first irradiated with LED at the indicated wavelengths (365 nm violet, 530 nm green) and then mechanically stimulated (cell poking) by a series of downward movements of a blunt pipette (red arrow in inset) at a velocity of 1 μm ms⁻¹ (only one trace is shown, see Supplementary Fig. 3b for full protocol). **e** Average light- (violet) and poking-evoked (black) current densities (mean ± s.e.m., number of cells is indicated in parentheses) obtained for each construct. Current noise limit (2σ) is indicated for each stimulus. Kruskal-Wallis followed by Dunn's post-test comparing to control mP1 data. *P* values (from top to bottom): *= 0.0147, ****<0.0001, ****<0.0001, *= 0.028, ****<0.0001. ***= 0.0002, and *= 0.0160. NT, non-transfected cells treated with MAT. Source data are provided as a Source Data file.

distinct genetic backgrounds[7,12,13,15]. However, it is unknown whether PIEZO, P2X, and ASIC share a common gating mechanism.

We, and others, have developed a series of photochemical optogenetics tools for P2X[18–21] and ASIC[21], in which the gating machinery has been reprogrammed to respond to light. These tools were based on a central photoisomerizable azobenzene, which can be toggled between a bent *cis*-isomer upon UV-irradiation and an elongated *trans*-isomer upon visible-light irradiation. Covalently attached to engineered cysteines introduced in the TM domain of the pore by site-directed mutagenesis, these photoswitches successfully provided control over channel gating of P2X and ASIC[18–21]. At first glance, extension of this method to PIEZO channels seems challenging because these proteins contain 114 TM helices, an exceptionally large number of TM helices compared to P2X and ASIC, which contain only 6-TM. However, a recent high-speed atomic force microscopy study has reported that it takes between 50 to 150 $k_B T$ (with $k_B$ = Boltzmann constant, and $T$ = temperature) to reversibly deform PIEZO1 structure from a curved, closed channel state into a flat, presumably open channel state[22] (Fig. 1b), a free energy difference ($\Delta G_O$) that is not far from the free energy difference between open and closed states of P2X (-12 $k_B T$)[23],

and between *cis* and *trans* configurations of a single azobenzene molecule ($\Delta G_{cis\text{-}trans}$ ~20 $k_B T$)[24], suggesting that azobenzene photo-isomerization energy could, in principle, be partially converted into PIEZO channel gating (Fig. 1c).

Here, we report a light-gated mouse PIEZO1 channel, whereby an azobenzene-based photoswitch covalently attached to an engineered cysteine residue in TM38, in a region that is spatially equivalent to that of P2X in TM2 which previously produced light activation with the same tethered photoswitch, rapidly triggers channel gating on light irradiation. Our results support that light-gated PIEZO1 recapitulates mechanically activated PIEZO1 biophysical properties, including single-channel conductance, inactivation, cation selectivity, and Yoda1 modulation.

## Results

### Engineering light-gated mP1 channels
We first computed a structural alignment between the central pore of mouse PIEZO1 (mP1) and the pore of rat P2X2 (rP2X2), and confirmed a close superposition of TM37 to TM1 and of TM38 to TM2 (Supplementary Fig. 1a, b). We focused on the mP1 region that is equivalent to

that containing the residue I328 from rP2X2, which was previously shown to endow robust light sensitivity to rP2X2 when mutated into cysteine and tethered to various chemical photoswitches[20], including maleimide ethylene azobenzene trimethyl ammonium[18], here renamed MAT (Fig. 1c and Supplementary Fig. 2). We found 6 residues from mP1 (S2211, L2212, G2463, Y2464, V2467, and G2468) to be spatially close to the equivalent region occupied by I328, and thus asked whether MAT tethering to these 6 residues, one-by-one, would confer light sensitivity to mP1 (Fig. 1a and Supplementary Fig. 1a, b).

We individually mutated selected mP1 residues into cysteine in the mP1-IRES-eGFP vector, and transiently transfected them in HEK-P1KO cells, a mechanically silent cell line that is void of endogenous PIEZO1 channels[25]. Twenty-four hours after transfection, cells were preincubated for 20 min with 200 μM MAT followed by extensive washing to remove unreacted MAT. We recorded patch-clamp currents from eGFP-positive cells in the whole-cell configuration while simultaneously illuminating the cell by switching light wavelength between 365 and 530 nm for 200 and 800 ms, respectively, via a LED irradiation system (Fig. 1d). As a control, a series of increasing mechanical stimulations (0.5–8 μm) was applied to the same cell by poking the cell with a piezo-electrically driven blunt glass pipette, as previously described[1].

No light-gated currents were recorded above current noise (2σ) in MAT-incubated non-transfected (NT) cells or in incubated cells expressing either mP1 or mutants S2211C, L2212C, G2463C, V2467C, and G2468C, while most of these mutants, including mP1, responded to poking stimulations with amplitudes that were similar to those recorded without MAT incubation (Fig. 1d, e and Supplementary Fig. 3a–d). However, a strong inward current was recorded for Y2464C at 365 nm that was not observed in the absence of MAT (Fig. 1d and Supplementary Fig. 3a, b). This light-gated current developed in $60 \pm 5$ ms (time-to-peak, mean ± s.e.m., $n = 15$ cells), appeared in the absence of external mechanical force, slightly inactivated during the 365-nm irradiation, and completely returned to the baseline in less than $461 \pm 13$ ms. Poking stimulation on the same cell induced a response that was of similar amplitude to that mechanically evoked on mP1 treated with MAT or the non-treated mutant (Fig. 1d, e and Supplementary Fig. 3a–e). On average, light-gated currents represented $5.3 \pm 0.7\%$ (mean ± s.e.m.) of maximal current density evoked by the largest poking stimulations (or $7.3 \pm 1.8\%$ of maximal current amplitude, Supplementary Fig. 3e), and were specific to cysteine engineering at position 2464, as no light-gated current was recorded in the isosteric mutant Y2464T that was otherwise activated by mechanical stimulations (Fig. 1e).

Although light-gated currents rapidly developed in 60 ms, they were ~12-times slower than those evoked by poking ($4.9 \pm 0.5$ ms, mean ± s.e.m., $n = 15$ cells), suggesting that bound MAT azobenzene isomerization induces a local force that is more slowly transduced to channel gating than the global force applied by cell poking. However, we found that decreasing light intensity of the 365-nm LED significantly increased the time to reach the light-gated current peak ($297 \pm 20$ ms), with a concomitant 2.3-fold decrease of current amplitude (Supplementary Fig. 4a–c), suggesting that light intensity tunes both activation kinetics and current magnitude.

We also found evidence for MAT labeling at two other cysteine positions: V2467C and G2463C. Instead of producing light-gated currents, labeling at these positions either reduced inactivation rates induced by continued poking stimulations for V2467C, a feature also observed for Y2464C (Supplementary Fig. 3f, g), or abolished the few poking-evoked currents that were recorded in the absence of MAT labeling for G2463C (Supplementary Fig. 3c, d). This last result suggests that the G2463C mutation, on its own, may already alter channel activity and/or cell trafficking. In the remaining text, we focus on MAT-labeled Y2464C, which we coined mOP1 (mouse OptoPIEZO1) hereafter.

## mOP1 recapitulates mP1 functional properties

We next determined the functional properties of mOP1 and compared them to those of mP1. We first tested the reproducibility of light-gated currents by illuminating cells with five cycles of constant irradiation (200 ms at 365 nm and 800 ms at 530 nm). As we had previously found that 200 ms of irradiation produced a slight current inactivation that may reduce subsequent responses (Fig. 1d), we found that two subsequent irradiation cycles had to be separated by at least a 60 s interval to obtain similar light-gated current amplitudes (Supplementary Fig. 4d, e). These data suggest that mOP1 inactivates and then recovers from channel inactivation, a phenomenon that is similar to mP1, for which a faster minimal 10 s time interval was needed for full recovery[26,27]. Using 60 s intervals, we recorded five cycles of stable light-gated currents that were time-locked to irradiations at 365 nm (Fig. 2a, b). No current was recorded at 530-nm light irradiation, consistent with a labeling in the dark-adapted trans configuration of MAT, and pre-irradiation at 530 nm had no impact on a subsequent light-gated current at 365 nm (cycle 1 after irradiation at 530-nm compared to cycle 1 control, Fig. 2a, b).

To further investigate mOP1 inactivation, we increased the duration of 365-nm illumination (from 200 ms to 1 s) while decreasing that of 530-nm (from 800 ms to 0 s) to maintain constant illumination time. Increasing 365-nm irradiation time induced a progressive decrease of subsequent light-gated currents that became significant at 800 ms (cycle 4), when compared to control currents recorded by constant 365-nm illumination (Fig. 2a, b). This inactivation process mirrors channel inactivation induced by prolonged mechanical stimulations of PIEZO channels[1]. Notably, inactivated currents recovered approximately two-fold after 2-min washing interspersed with a 1 s 530-nm light irradiation to convert the azobenzene in the activatable, trans configuration (Fig. 2c). In addition, we found a clear correlation between the time interval separating two subsequent light-gated responses and channel recovery (Supplementary Fig. 4d, e), as well as a clear relationship between the level of current inactivation induced by 365 nm light and the level of channel recovery (Supplementary Fig. 4f, g). Altogether, these data suggest that mOP1 reversibly recovers from inactivation, a process that is similar to that observed for mP1[1].

Investigating the kinetics of light-induced, current-fading transitions, we found that the time constant values of current inactivation ($\tau_{inact}$) evoked by a long irradiation at 365 nm were significantly higher than those of current deactivation ($\tau_{deact}$) evoked at 530 nm light following a short (50 ms) irradiation at 365 nm, suggesting that the transition towards inactivation induced by cis-MAT is slower than the backward open-closed transition induced by trans-MAT (Fig. 2d, e). This backward transition is reminiscent of PIEZO1 current deactivation which occurs after the removal of a very brief mechanical stimulation[26]. However, we found that $\tau_{inact}$ values of these light-gated transitions were considerably higher (38-fold) than those evoked mechanically by poking (Fig. 2e and Supplementary Fig. 3g), suggesting that channel inactivation induced by light is slower than that evoked by mechanical stimulation, a result that mirrored activation kinetics. Likewise, decreasing light intensity of the 365-nm LED significantly increased $\tau_{inact}$ values, suggesting that light-induced channel inactivation also depends on light intensity (Supplementary Fig. 4a, c). We do not exclude the possibility that these light-gated transitions might contain more complex kinetics that was not resolved here.

Next, we recorded light-gated currents of mOP1 at different holding potentials and showed a current-voltage relationship that was similar to that of PIEZO1 (ref. [28]), including the voltage-dependent inactivation (current inactivation was less pronounced or absent at positive holding potentials) (Fig. 2f, g, black traces). No light-gated currents were recorded at any voltages in MAT-incubated NT cells. To exclude a possible ion selectivity switch, from cation to anion, due to the presence of the MAT positive charge, we measured the reversal potential ($E_{rev}$) in different external solutions, in which either external

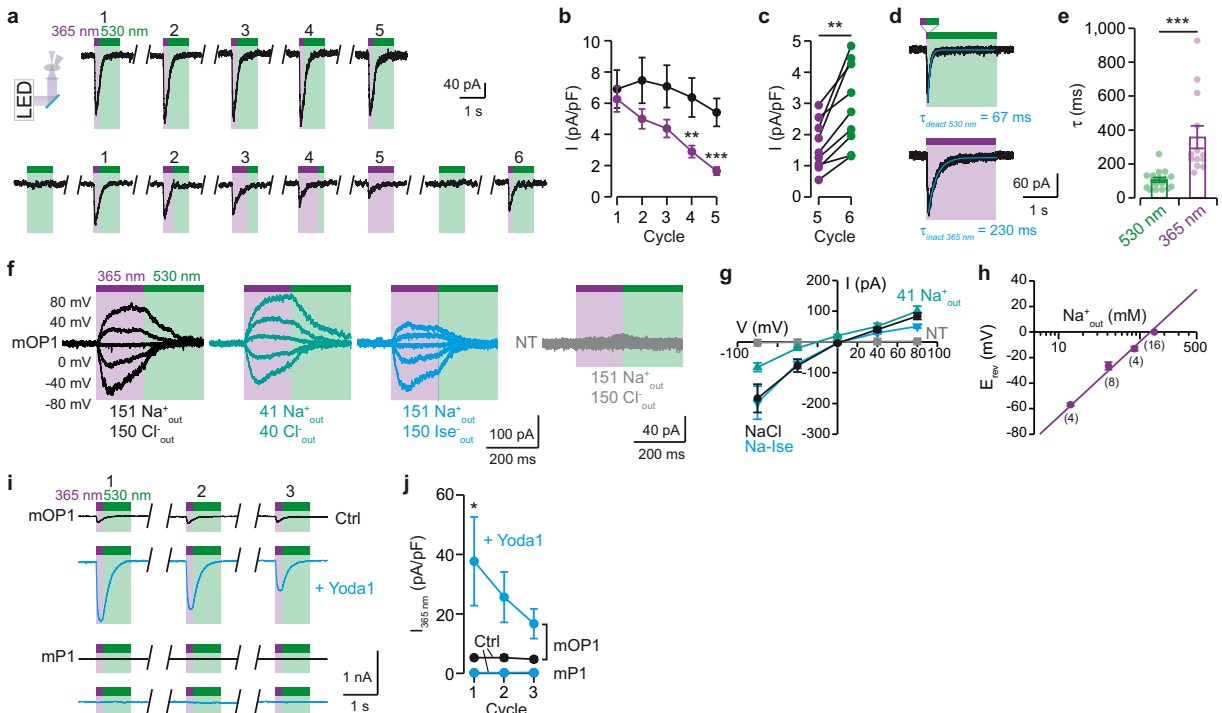

**Fig. 2 | mOP1 recapitulates mP1 functional properties. a** Representative traces of whole-cell currents at −80 mV of mOP1 from two different cells in response to several cycles of constant (upper trace) or varying (bottom trace) irradiation time. Cycles were separated by 1 min. **b** Average current density as a function of irradiation cycle (constant time, black data; varying time, violet data, n = 9 cells per each irradiation condition). Comparison with two-tailed Mann–Whitney in between groups. P values: **= 0.004, ***= 0.0008. **c** Individual current density values between cycles 5 and 6. Two-tailed paired t test. P value: **= 0.0013. **d** Current deactivation elicited by 530-nm light following 50 ms activation at 365 nm (top) and current inactivation elicited by 2-s activation at 365 nm (bottom). Fading currents were fitted with an exponential equation giving $\tau$ values. **e** Average $\tau$ values (n = 13 cells for 365 nm and 17 cells for 530 nm). Two-tailed Mann–Whitney test, P value: ***<0.0001. **f** Whole-cell light-gated currents recorded at holding potentials ranging

from −80 mV to +80 mV in the indicated extracellular solutions. NT, non-transfected cells treated with MAT. **g** Average current-voltage relationships of light-gated currents in indicated extracellular solutions (n = 6 cells for NT, 15 cells for NaCl, 7 cells for Na-Ise, and 8 cells for 41 $Na^+_{out}$). **h** Corresponding $E_{rev}$ values as a function of external $Na^+$ concentration (number of cells is indicated in parentheses) fitted to a simplified Goldmann-Hodgkin-Katz equation with no chloride permeability (see Methods). **i** Bath application of 10 µM Yoda1 (cyan traces) potentiated light-gated currents of mOP1 compared to control (black traces), but had no effect in light-exposed MAT-treated cells expressing mP1. **j** Average light-evoked current density (n = 6 cells for mOP1 and 5 cells for mP1). Comparison with two-tailed Wilcoxon signed rank test. P value: *= 0.0313. All data are presented as mean ± s.e.m. Source data are provided as a Source Data file.

$Cl^-$ was replaced by isethionate or external $Na^+$ was reduced. We found evidence that mOP1 remains cation-selective ($P_{Na}/P_{Cs} = 1.15 ± 0.07$, n = 4–8), with no evidence of chloride permeability (Fig. 2f–h).

Last, we found that bath application of 10 µM of the positive modulator Yoda1 strongly potentiated mOP1 light-gated currents (7.2 ± 2.3-fold, n = 6 cells, two-tailed Wilcoxon paired test, P = 0.0313), a feature that was also observed for mP1 mechanical stimulations[9]. As a control, no light-gated currents were recorded for mP1 (Fig. 2i, j), although we verified that Yoda1 perfusion did increase holding currents of mP1 as a proof of Yoda1 activity (−205 ± 37 pA with Yoda1, −138 ± 33 pA without Yoda1; mean ± s.e.m., n = 5 cells, P = 0.0497).

Altogether, these data demonstrate that light-gated currents of mOP1 feature the same functional properties as mP1, including channel inactivation, cation permeation, voltage dependency, and sensitivity to a modulating agent.

## mOP1 recapitulates features of selected PIEZO1 mutations

To further validate mOP1 application, we next sought to establish whether mOP1 can distinguish PIEZO1 mutations that were previously shown to alter channel biophysical properties. We focused on the single R2482H and triple E2257K/E2258K/D2264K mutations, which were previously described as slower mechanical-evoked inactivation and deactivation[26,29], and asked whether light can reproduce these mechanical-induced features. We, therefore, introduced these mutations into the Y2464C background (R2482H mOP1 and E2257K/E2258K/D2264K mOP1, named hereafter KKK mOP1) and labeled

mutant-expressing cells with MAT. We found that light significantly increased both $\tau_{inact}$ at 365 nm and $\tau_{deact}$ at 530 nm in the two mutants, when compared to mOP1, in a manner that was similar to poking stimulations and published data[26,29] (Fig. 3a, b). These results, therefore, suggest that light-induced azobenzene isomerization recapitulates inactivation and deactivation kinetics induced by mechanical stimulations in PIEZO1 mutants.

We also found that light-gated currents significantly developed more slowly for both mutants, whereas no change was observed for mechanical stimulations (although KKK mOP1 currents tended to develop more slowly upon mechanical activation, P = 0.0932, Fig. 3b). In addition, the presence of both mutations significantly increased light-gated current amplitudes, as compared to poking-evoked current amplitudes which remained unchanged (Fig. 3b). As a consequence, light-gated currents represented ~40% of maximal poking-induced currents in both mutants (38 ± 6% for R2482H mOP1 and 43 ± 5% for KKK mOP1), indicating that slowing inactivation kinetics increases light-gated current amplitudes.

## Simplicity of mOP1 use in cell imaging

Having shown that mOP1 recapitulates mP1 functional properties, including selected PIEZO1 mutants, we next used $Ca^{2+}$ imaging to validate the easy handling of mOP1. We took advantage of the fact that mP1 is selective to $Ca^{2+}$ (ref. [1]) to replace eGFP in the mP1-IRES-eGFP vector with the genetically encoded fluorescent $Ca^{2+}$ indicator GCaMP6, which reports $Ca^{2+}$ influx in cells bathed with $Ca^{2+}$-containing

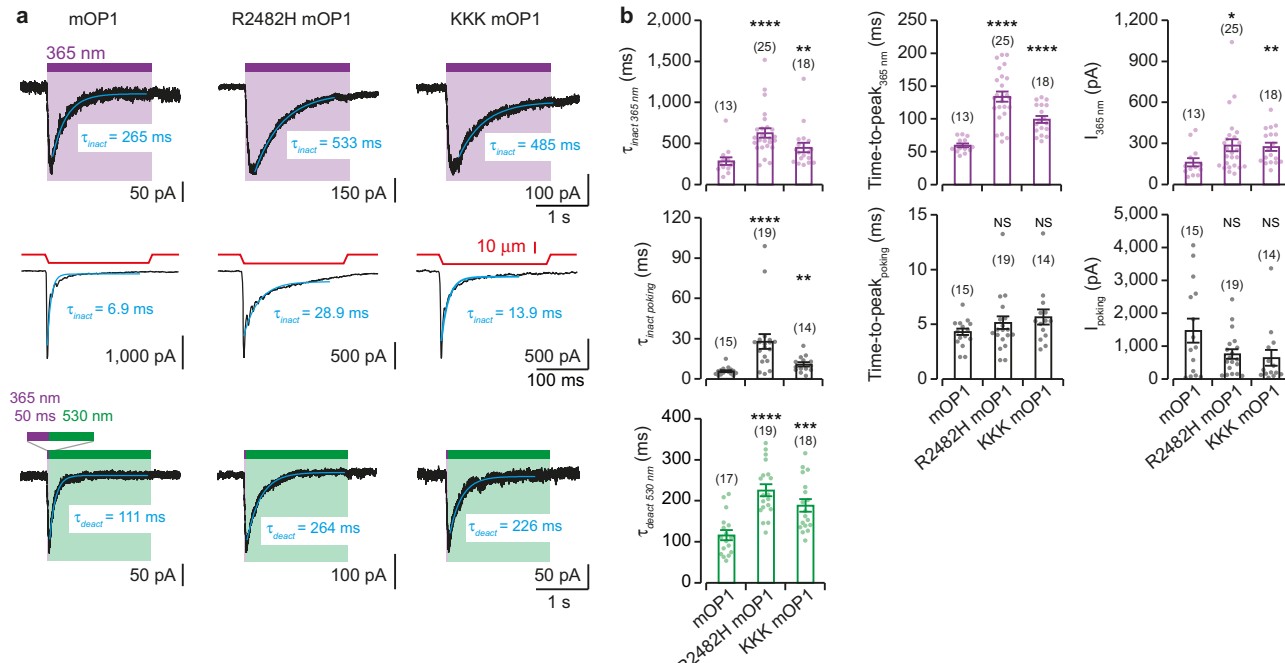

**Fig. 3 | mOP1 recapitulates functional properties of selected PIEZO1 mutants.** **a** Whole-cell current inactivation at −80 mV elicited by 2-s activation at 365 nm (top) or 200-ms by cell poking (middle) and current deactivation (bottom) elicited by 530-nm light following 50 ms activation at 365 nm in cells expressing either mOP1, R2482H mOP1, or E2257K/E2258K/D2264K mOP1 (KKK mOP1). Fading currents were fitted with an exponential equation giving $\tau$ values. **b** Average $\tau$, time-to-peak, and current amplitude values for 365-nm light (top), cell poking (middle), and 530-nm light (bottom) conditions (number of cells is indicated in parentheses) for mOP1 (left), R2482H mOP1 (middle) and KKK mOP1 (right). Comparison to control mOP1 with two-tailed Mann–Whitney test or two-tailed unpaired $t$ test (for time-to-peak$_{365\,nm}$ and $\tau_{deact\,530\,nm}$). $P$ values: ****<0.0001, ***= 0.0007, ** = 0.0049 for $\tau_{inact\,365\,nm}$ and $I_{365\,nm}$, **= 0.0051 for $\tau_{inact\,poking}$, *= 0.0196 and NS = 0.3538, 0.0932, 0.3361, and 0.2340 (from left to right). NS, not significant. All data are presented as mean ± s.e.m. Source data are provided as a Source Data file.

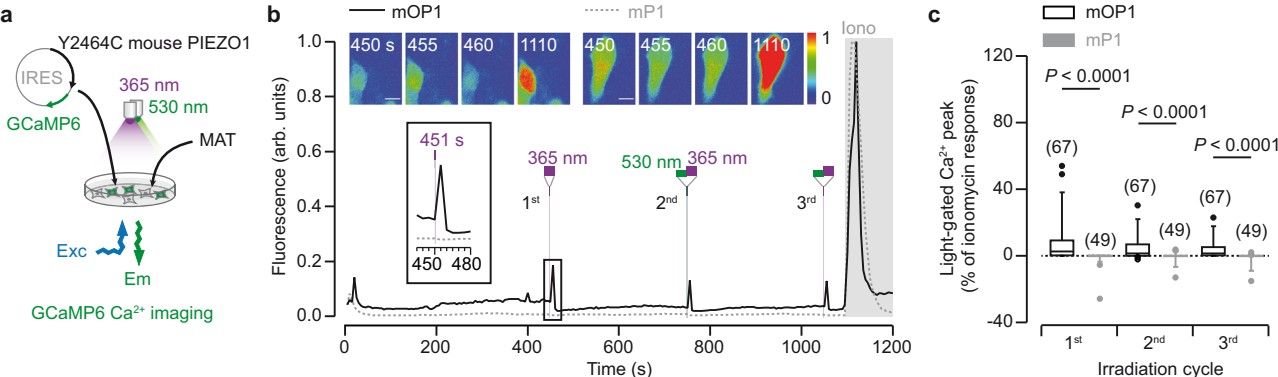

**Fig. 4 | mOP1 allows to interrogate PIEZO1 function in cell imaging.** **a** Schematic representation of GCaMP6 Ca²⁺ imaging following mOP1 activation. **b** Time-lapse of fluorescence emission at 510 nm (arbitrary units) in response to 1 s irradiations at 365-nm in MAT-treated cells expressing either mOP1 (thick black trace) or mP1 (dotted gray trace). Note 1-s pre-irradiations at 530 nm just before the second and third irradiation at 365 nm. Ionomycin (5 μM) was applied as a control (gray shadow). Corresponding fluorescent images are shown on the top at the indicated times. Scale bar, 10 μm. Inset, zoom of the first irradiation. **c** Box plot of light-gated Ca²⁺ peak responses normalized to that of ionomycin in the function of irradiation cycles. Responses are presented as median (center), 25–75 percentile (box), and 5–95 percentile (whisker). Outliers extending beyond whiskers (1.25× whisker length) are shown. Indicated $P$ values are from two-tailed Mann–Whitney test. Number of cells is indicated in parentheses. Source data are provided as a Source Data file.

buffers (Fig. 4a). Following MAT labeling and extensive washout, a sharp light-induced Ca²⁺ response at 365 nm was specifically detected in cells expressing mOP1, but not in cells expressing mP1, for which no signal was detected (Fig. 4b, c). This Ca²⁺ response was reproducibly detected each time cells were illuminated at 365-nm light, provided the azobenzene was formerly switched back to the *trans* configuration. On average, the first light-gated responses represented 7.1 ± 1.4% ($n$ = 67 cells) of control responses elicited by the Ca²⁺ ionophore ionomycin that was applied at the end of the experiment (Fig. 4b). When normalized to Yoda1, the specific PIEZO1 activator commonly used in cell

imaging assays[11,30,31], these responses increased to 15.9 ± 4.5% ($n$ = 33 cells), whereas no response was detected in the absence of MAT (Supplementary Fig. 5a, c).

Bath application of 10 μM Yoda1 induced a strong and transient rise of intracellular Ca²⁺ in both mP1 and mOP1 that decreased to a plateau of relatively high activity (between 30 and 50% of the ionomycin control) (Supplementary Fig. 5b). This plateau was expected because of the continuous bath presence of Yoda1 that constitutively gated PIEZO activity (there were no washing steps allowing Yoda1 removal for all experiments). In line with the fact that the MAT-

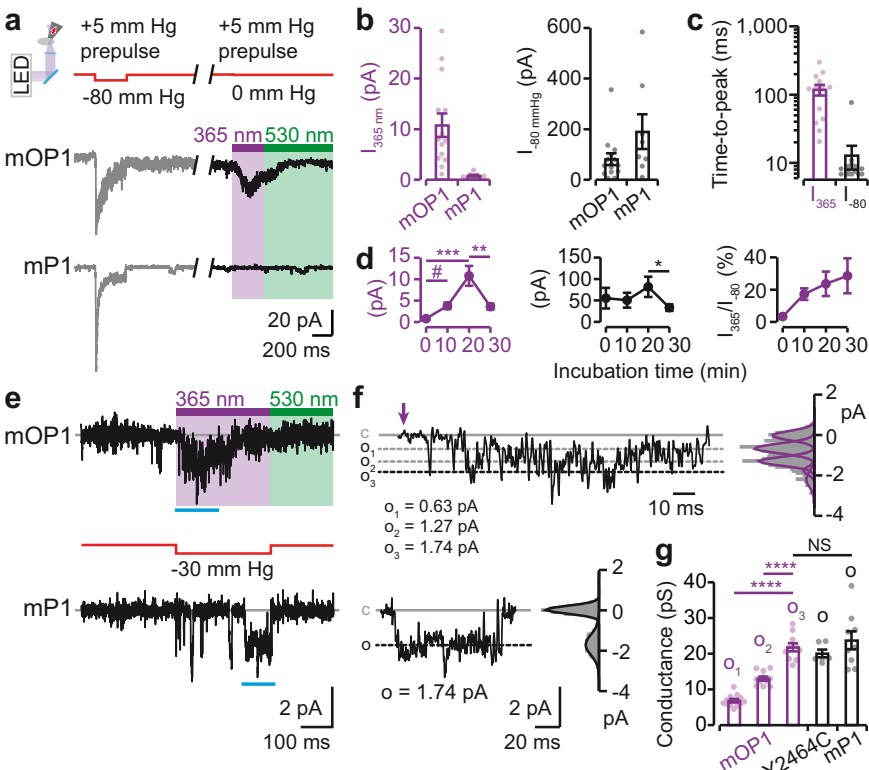

**Fig. 5 | Single-channel currents of mOP1. a** Typical inward currents at −80 mV in the cell-attached configuration evoked by first applying negative pressure through the recording pipette (red arrow in inset) and then light irradiation 10 s after. Shown are two patches from MAT-treated cells expressing either mOP1 (upper traces) or mP1 (bottom traces). A pre-pulse of +5 mm Hg was applied to the patch to minimize resting membrane tension, as previously described[32]. **b** Average light-gated (violet histograms) and pressure-evoked (black histograms) currents on mOP1 and mP1 (n = 14 cells for mOP1 and 8 for mP1). **c** Average time-to-peak activation data on mOP1 (n = 14 cells for each condition). **d** Current dependency as a function of incubation time of cells with 200 μM MAT (n = 8, 16, 14, and 15 cells for incubation times 0, 10, 20, and 30 min, respectively). Comparison with two-tailed Mann–Whitney test. P values: *=0.0283, #=0.0011, **=0.0047, ***=0.0001. **e** Typical single-channel currents at −80 mV in the cell-attached configuration elicited by light irradiation or pressure stimulation of cells expressing either mOP1 (MAT-

treated) or mP1 (not treated). Channel openings are downward deflections. **f** Sections of the recordings shown in **e** (cyan lines below traces) are displayed at higher time resolutions. Labels c and o denote, respectively, closed and open conductance levels of mP1 and $o_1$, $o_2$ and $o_3$ represent the three open conductance levels of mOP1. The violet arrow indicates the start of irradiation. Corresponding all-point histograms with Gaussian fits are shown right of the traces. Data were filtered at 1 kHz. **g** Average single-channel conductance determined at −80 mV (n = 15 cell-attached patches for mOP1, 6 for Y2464C (MAT not treated), and nine for mP1) for light (violet data) and pressure (black data) stimulations. Comparison with two-tailed unpaired t test. P values: ****<0.0001. NS, not significant (P = 0.4817 between $o_3$ and o mP1). Comparison with two-tailed Mann–Whitney test for Y2464C data. P = 0.5622 between $o_3$ and o Y2464C, and P = 0.5287 between o Y2464C and o mP1). All data are presented as mean ± s.e.m. Source data are provided as a Source Data file.

unlabeled Y2464C mutant inactivated more rapidly than mP1 (Supplementary Fig. 3f, g), the decrease of fluorescent signal was also more rapid for mOP1 than for mP1. These data thus suggest that most Yoda1-induced $Ca^{2+}$ responses come from unlabeled channels. In support of this hypothesis, mOP1 photoactivation induced sharp and reproducible $Ca^{2+}$ responses in the presence of Yoda1 that amounted to 26.3 ± 3.2% (n = 48 cells) of the ionomycin signal, in line with the potentiating effect of Yoda1 in light-gated currents (Supplementary Fig. 5b). Surprisingly, significant light-gated responses were also detected in MAT-treated cells expressing mP1 (3.1 ± 0.5% of ionomycin signal, n = 108 cells); however, these signals appeared to be independent of MAT as they were observed in the absence of labeling in both Y2464C and mP1, and only in the presence of Yoda1 (Supplementary Fig. 5c).

Altogether, these data demonstrate that mOP1 can be used to simply interrogate PIEZO function in cell imaging systems, paving the way for future PIEZO investigations, including high-throughput screening drug discovery.

### Three unitary conductance levels of mOP1
We next investigated the mechanism by which light activates mOP1 and compared light-gated currents to mechanically-activated

currents. We switched to the cell-attached mode because it allowed recording channel activity that was only present in the patch membrane, thus yielding direct and reliable comparisons between light- and mechanically-activated currents, which are more difficult to achieve with poking stimulations in the whole-cell configuration. We recorded and compared, in the same patch, light-gated mOP1 currents to those evoked by applying negative pressure through the recording pipette (−80 mm Hg for 300 ms). Each patch was first stretched by a negative pressure pulse followed 10 s after by one cycle of illumination (365-nm for 300 ms and 530-nm for 700 ms) (Fig. 5a). To minimize resting membrane tension, a pre-pulse of +5 mm Hg was applied to the patch for 5 s prior to light or mechanical stimulations, as previously described[32] (Fig. 5a and Supplementary Fig. 6a).

Consistent with whole-cell data, patches from cells expressing mOP1, but not those expressing mP1, showed robust light-gated currents at 365-nm that returned at 530-nm light (Fig. 5a, b). These light-gated currents developed in 118 ± 21 ms (time-to-peak, mean ± s.e.m., n = 14 cells) and amounted to 23.6 ± 7.6% of those evoked by near maximal pressure (Fig. 5c, d and Supplementary Fig. 6b). As already found in whole-cell recordings, light-gated currents activated 13.7 ± 3.0-times slower than pressure-evoked currents recorded in the

same patches (Fig. 5c), confirming slower mechanistic processes for light than for mechanical stimulations.

Varying MAT incubation times revealed optimal light-gated currents at 20 min incubation (Fig. 5d). Indeed, although light-gated current yield (ratio of light- to pressure-gated currents) was higher at 30 min than at 20 min, an unexpected decrease of both light- and pressure-gated currents was observed at 30 min. This reduction of activity was not related to a change of PIEZO1 plasma membrane expression, as assessed by cell-surface protein biotinylation (Supplementary Fig. 7), suggesting that long incubation times >20 min might induce a reduction of PIEZO1 gating through an unknown mechanism. However, MAT incubation at 20 min has no effect on pressure responses on both mP1 and mOP1, neither on maximal current ($I_{max}$) nor on pressure values giving a half-maximal current ($P_{50}$), suggesting that MAT labeling for 20 min did not change any functional pressure-evoked properties, although the Y2464C mutant on its own (irrespective of labeling) has a decreased $I_{max}$ efficacy compared with mP1 (Supplementary Fig. 6b–d).

We next recorded single-channel currents in patches from cells expressing mOP1. Light-evoked unitary currents operated in a highly flickering mode of activity, compared with currents induced by pressure on mP1 (Fig. 5e). These light-gated currents were nevertheless resolved in three distinct conductance levels of $6.8 \pm 0.5$ pS, $13.0 \pm 0.6$ pS and $21.8 \pm 1.2$ pS ($n = 15$ cell-attached patches analyzed separately), named, respectively, $o_1$, $o_2$, and $o_3$, and were linearly proportional with labeled subunits (Fig. 5f and Supplementary Fig. 6e, f). Because the simultaneous presence of these distinct conductance levels was not systematically detected in every single patch, we further performed event analysis from all 15 cell-attached patches (1,002 events analyzed) and consistently revealed three distinct conductance levels that were very similar to those individually determined from each patch (Supplementary Fig. 6g). $O_1$ and $o_2$ were significantly different from $o_3$, while $o_3$ was not different from the conductance level (o) of unlabeled mP1 or Y2464C stimulated with pressure (Fig. 5g and Supplementary Fig. 6h), suggesting that $o_3$ would correspond to full channel opening with a single-channel conductance that was not far from those previously reported[2,7]. Consistent with random labeling of one, two, or three subunits per channel, our data support the hypothesis that $o_1$, $o_2$, and $o_3$ would correspond to a gradual pore opening induced, respectively, by one, two, and three MAT molecules.

## Molecular motions in mOP1 gating

To investigate light-gated molecular motions, we next asked whether the activation mechanism induced by light was similar or not to that evoked by mechanical force. To tackle this challenging question, we took advantage of a recent engineering work showing that locking the dome-shaped structure of mP1 by spontaneous cross-linking of E2257C from the cap with R1762C from the blade abolished mechanical activation, and that mechanical activation can be reversibly restored by selectively reducing the engineered disulfide link with 1,4-dithiothreitol (DTT), thus releasing structural constrains[26] (Fig. 6a). We, therefore, asked whether mOP1 also obeys this logical scheme, and introduced the Y2464C mutation into the double cysteine mutant R1762C/E2257C background. Cells transiently transfected with the triple mutant R1762C/E2257C/Y2464C labeled with MAT (C/C mOP1) responded to 365-nm light irradiation only in the presence of DTT that was alternatively applied between nominally DTT-free buffer (Fig. 6b, c). As controls, no light-gated currents were observed in R1762C/E2257C mP1 (C/C mP1) incubated with MAT, both in the presence or absence of DTT (Fig. 6b, c), and robust poking-evoked currents were recorded in the presence of DTT in R1762C/E2257C/Y2464C with inactivation kinetics that depended on MAT labeling (Supplementary Fig. 8), as observed for the Y2464C mutation (Supplementary Fig. 3g). These data thus demonstrate that azobenzene photoisomerization induces a molecular gating motion that requires full detachment of the blades and cap, a mechanism that is similar to mechanical activation[26].

## The blades contribute to the rapidity of light-induced channel opening

To further investigate the mechanism by which photoisomerization of covalently bound MAT produces channel opening, we modeled MAT molecules in the recently determined cryo-electron microscopy (cryo-EM) structures of mP1 resolved in a curved, closed state and in a flattened, presumably open state[17]. We introduced the Y2464C mutation and docked MAT molecules, in the *trans* and *cis* configurations, at distances compatible with a covalent bond between the sulfur atom of the engineered cysteine and one of the carbon atoms of the maleimide, both in curved and flattened structures. In the curved state, our docking modeling revealed a large majority of *cis*-MAT poses in a region supposedly occupied by membrane lipids, whereas *trans*-MAT poses were found in close proximity to the protein (Supplementary Fig. 9a). These results suggest that MAT *trans-cis* photoisomerization may directly push on surrounding phospholipids, producing a local membrane perturbation that can be sensed by the neighbored blades which eventually gate the channel. In the flattened structure, however, our docking simulations revealed another plausible mechanism, in which *cis*-MAT poses were found to fill the large gap that was present between two adjacent TM38 helices (Supplementary Fig. 9b). Because *trans*-MAT poses did not span this gap, our data suggest that *cis*-MAT may also stabilize the flattened, presumably open state of PIEZO channels.

To discriminate between these two possibilities, yet complementary, mechanisms we asked whether removing the force-sensor blades from mP1 would provide a channel that can still be gated by light. We thus designed a truncated version of mP1 (Δblade mP1, starting from residue E2172, Supplementary Fig. 10) previously shown to be expressed as a mechanically insensitive, ion-conducting pore displaying spontaneous activity[28], introduced the Y2464C mutation on this background, and labeled it with MAT (named Δblade mOP1, Fig. 6d). To detect spontaneous pore activity, we used gadolinium ($Gd^{3+}$) and ruthenium red (RR), two known mP1 pore blockers[1], and found a specific reduction of the holding current upon blocker application that was not observed in non-transfected cells (Fig. 6e and Supplementary Fig. 11). Notably, MAT-treated cells expressing Δblade mOP1, but not those expressing Δblade mP1, were still able to respond to 365-nm light, clearly showing that the three blades of mOP1 are unnecessary for light-gated motions (Fig. 6e, f). These data thus rather support a mechanism by which covalently bound *cis*-MAT molecules could insert between pore-lining TM helices to stabilize an open conformation. The occurrence of spontaneous channel activity may favor such insertion.

However, our data from this truncated protein also revealed the importance of the blades in several features. First, channel opening evoked by *cis*-MAT was ~70-times slower in Δblade mOP1 than in the full-length mOP1 (Fig. 6g, h), indicating that the blades, accounting for >85% of the PIEZO1 primary structure, critically contribute to the rapidity of channel opening. Second, light-gated currents are poorly deactivated at 530 nm (Fig. 6e), suggesting that the blades contribute to channel deactivation. Third, no channel inactivation was noticed in Δblade mOP1, even after a 5 s long irradiation (Fig. 6e), demonstrating that the three blades of mOP1 control channel inactivation.

## Discussion

We report mOP1 as a powerful self-operating, light-gated PIEZO1 channel that can be easily used to specifically interrogate PIEZO1 function. We show that mOP1 retains the hallmark features of PIEZO1 channels, including cation selectivity, voltage dependency, Yoda1 potentiation, and channel inactivation and recovery processes. We further show that light-induced azobenzene isomerization

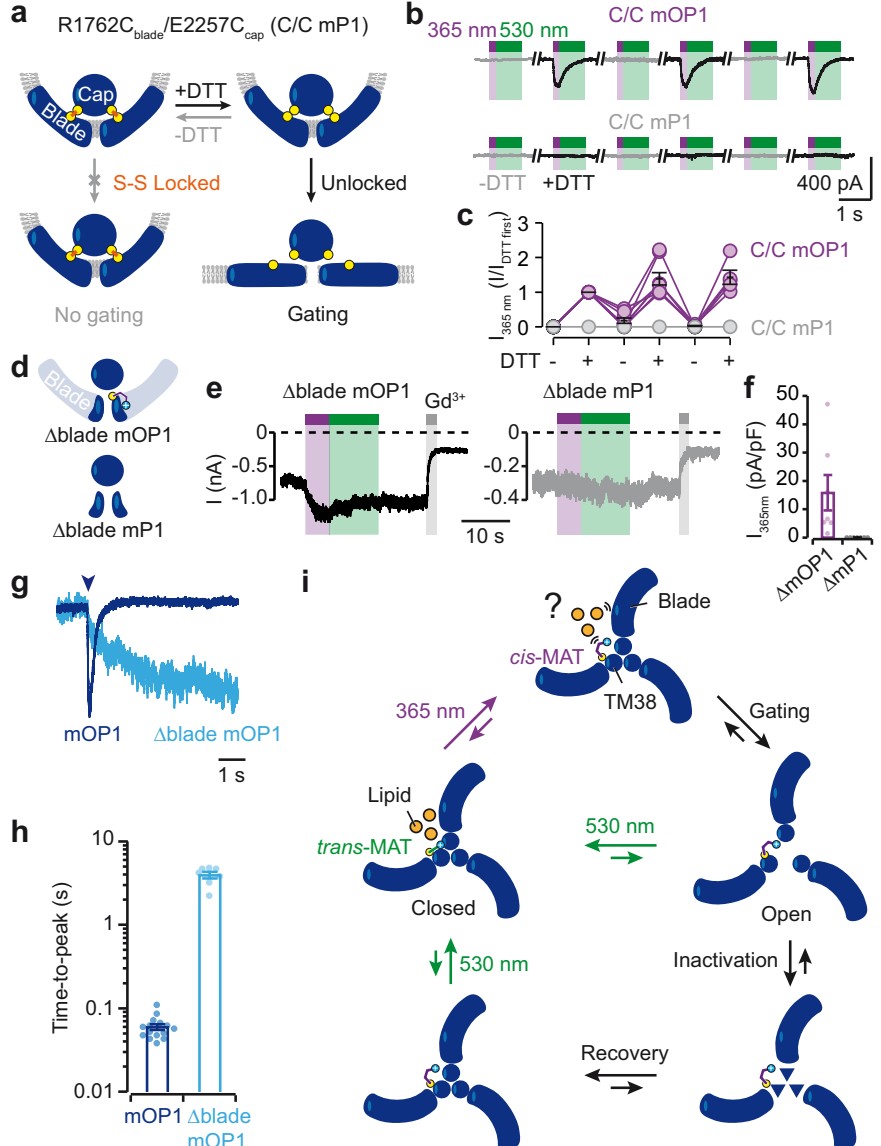

**Fig. 6 | Light-gated molecular mechanism in mOP1. a** Cartoon depicting the strategy of gating control of mOP1 by engineering spontaneous disulfide bonds between R1762C from the blade and E2257C from the cap. **b**, Representative light-gated inward currents at −80 mV (365 nm violet and 530 nm green) in MAT-treated cells expressing either the triple mutant R1762C/E2257C/Y2464C (C/C mOP1) or the double mutant R1762C/E2257C (C/C mP1) in the absence (gray traces) or presence (black traces) of 10 mM DTT. Currents were recorded 2.5-min after DTT application or 5-min after DTT washings. **c** Average light-evoked currents (normalized to the first application of DTT) recorded in the absence or presence of DTT, for C/C mOP1 (violet circles) and C/C mP1 (gray circles) ($n = 6$ cells for C/C mP1 and nine cells for C/C mOP1). **d** Cartoon showing Δblade mOP1 and Δblade mP1 (not carrying the Y2464C mutation). **e** Representative whole-cell currents at −80 mV of Δblade mOP1 (left) and Δblade mP1 (right) in response to light irradiation from cells treated with MAT. Note the presence of leaky currents that were reduced upon 30 μM Gd$^{3+}$ application demonstrating spontaneously opened channels. **f** Average current density obtained at 365 nm for Δblade mOP1 (ΔmOP1) and Δblade mP1 (ΔmP1) ($n = 7$ cells). **g** Superimposed light-gated currents of mOP1 (blue trace) and Δblade mOP1 (cyan trace) time-locked to light activation (arrowhead). **h** Corresponding time-to-peak values ($n = 15$ cells for mOP1 and 7 for Δblade mOP1). **i** Schematic representation of light-induced gating mechanism in mOP1. Due to the occurrence of photostationary states, conversion yields of light-induced transitions (violet or green arrows) are not total. For clarity, only one labeled MAT molecule is shown. All data are presented as mean ± s.e.m. Source data are provided as a Source Data file.

recapitulates mechanically induced biophysical features of two previously described PIEZO1 mutations, including the gain-of-function R2482H[29], for which the equivalent human R2456H mutation causes hereditary xerocytosis[33–35].

Our data also reveal some limitations of mOP1. First, whole-cell light-gated currents represent only a fraction (between 5 and 7%) of those mechanically activated by cell poking and it is unknown if this amount is sufficient to trigger physiological responses. Although we present evidence that light activation is sufficient to produce significant increases in intracellular Ca$^{2+}$, it remains to be determined whether these low currents might be sufficient to trigger other

signaling pathways. However, we found that light-gated current yield increased to ~25% in cell-attached patches when currents were normalized to those induced by pressure. Therefore, the actual yield of light-gated activation is difficult to assess with high precision and seems to depend on the mechanical stimulation mode. In support of this hypothesis, it has been found that PIEZO1 can mediate both localized and whole-cell mechanical responses via two different gating models, the force-from-lipids and the force-from-filaments models[36–38]. Although the relative contribution of these gating models to PIEZO1 activation is unclear in HEK cells, they might explain the difference in the light-gated yield values. In addition, we found that this

yield can be increased up to ~40% of mechanically activated whole-cell currents in the gain-of-function R2482H, thus suggesting that light-gated activation might be sufficient to trigger, at least partially, physiological responses in mice carrying this mutation[29,39]. The second limitation of mOP1 is the relatively slow light-induced transitions as compared to faster mechanically induced processes. One possibility is that different gating mechanisms are at play. However, this hypothesis seems unlikely as we found that decreasing light intensity tuned down both light-induced activation and inactivation kinetics, as well as current amplitude. Our data, therefore, support the idea that light-gated transitions can be accelerated to rates approaching those induced by mechanical stimulations by using more powerful irradiation devices. This may also increase the light-induced current amplitude.

We provide evidence that the blades are not necessary for light-gated motions, but they are critically involved in the rapidity of light-induced channel opening. It is unclear how the blades accelerate light-induced channel opening, but it might be speculated that *trans*-to-*cis* azobenzene photoisomerization can induce a local membrane tension perturbation that is rapidly sensed by the blades, which, in turn, respond by triggering channel opening (Fig. 6i). This rapid force-from-azobenzene gating mechanism would thus be reminiscent of the force-from-lipids gating mechanism of PIEZO1 (refs. [36,40]), and fits with the exquisite sensitivity of these channels to detect very small membrane tension perturbations[41,42]. Once activated, the open pore is transiently stabilized by the *cis*-isomer before entering in inactivation followed by recovery.

Our results push the limits of azobenzene-based methods to unusually large ion channels. It has been shown that the free energy difference between closed and open conformations of PIEZO1 embedded in a membrane with lateral membrane tension γ, can be defined as

$$\triangle G = \triangle G_O - \gamma \triangle A \tag{1}$$

where $\Delta G_O$ is the free energy difference at rest (i.e., at zero membrane tension) and $\gamma \Delta A$ is the external mechanical energy that is needed to open the channel ($\Delta A$ is the area membrane expansion associated with channel opening)[13]. For mP1, $\Delta G_O$ was estimated to lie from 2–10 $k_B T$ in cellular membranes[32,36,40] to 50–150 $k_B T$ in reconstituted membranes[22]. Considering $\Delta G_{cis\text{-}trans}$ is ~20 $k_B T$ for one photoswitchable molecule, azobenzene photoisomerization, therefore, provides sufficient external energy to substitute for mechanical forces. Given the reversibility of azobenzene photoisomerization, our data thus opens new avenues to extend our technology to other mechanically activated ion channels.

The mechanism by which covalently bound MAT produces light-gated currents needs further experimental testing, but seems to differ from that operating for P2X and ASIC, for which channel opening occurred in the *trans* configuration of MAT or of other chemically similar photoswitches[18-21]. Recent structural data have revealed the dynamic outward motion of trimeric pore-lining TM helices during channel gating[17,43-45], especially on the extracellular side of these helices, where chemical photoswitches have been covalently tethered to lend control over channel opening. Therefore, it seems that this dynamic region localized at the subunit interface represents a pivotal point in trimeric ion channels for the effective manipulation of channel gating with chemical actuators.

## Methods
### Chemical synthesis of MAT
MAT (maleimide ethylene azobenzene trimethyl ammonium) was resynthesized as previously described[18]. Briefly, 4-[(*E*)−2-(4-amino-phenyl)diazen-1yl]aniline (500 mg, 2.35 mmol) was desymmetrized by the coupling of (3-(2,5-dioxo-2,5-dihydro-1*H*-pyrrol-1-yl)propanoic acid (438.46 mg, 2.83 mmol) using O-Benzotriazole-*N,N,N′,N′*-tetramethyl-

uronium-hexafluoro-phosphate (HBTU) (1.072 g, 2.83 mmol) in presence of triethylamine (286.05 mg, 0.394 mL, 2.83 mmol) in 15 mL of dry acetonitrile under argon atmosphere at 20 °C for 16 h under magnetic stirring. After solvent evaporation, the mono-coupled product was extracted 3 times with AcOEt and purified by flash chromatography (Heptane: AcOEt gradient from 1:1 to 1:4) to yield an orange solid (335.37 mg, 0.96 mmol, 49%) that was then reacted 16 h at 20 °C under magnetic stirring in 15 mL of dry DMF with N,N-Diisopropyl-lethylamine (357.87 mg, 0.482 mL, 2.87 mmol) in presence of the acyl chloride generated by the reaction of oxalyl chloride (193.30 mg, 0.130 mL, 1.52 mmol) on 1-carboxy-*N,N,N*-trimethylmethanaminium chloride (233.93 mg, 1.52 mmol) in 10 mL of dry DMF under argon atmosphere for 2 h at 20 °C under magnetic stirring. After solvent evaporation, compound was purified by reverse phase HPLC (H₂O 0.1% TFA: acetonitrile, gradient from 1:0 to 0:1 over 30 min) to yield the desired product (213.59 mg, 0.46 mmol, 48%), as confirmed by NMR spectroscopy (Supplementary Fig. 2).

¹H-NMR (400 MHz, D₂O): δ 7.87 (dd, *J* = 12.9, 8.7 Hz, 4H), 7.71 (d, *J* = 8.7 Hz, 2H), 7.59 (d, *J* = 8.7 Hz, 2H), 6.86 (s, 2H), 4.33 (s, 2H), 3.90 (t, *J* = 6.5 Hz, 2H), 3.40 (s, 9H), 2.71 (t, *J* = 6.5 Hz, 2H). HRMS (ESI) (*m/z*): Exact mass calculated for C₂₄H₂₇N₆O₄ [M]⁺: 463.2088; found, 463. 2115.

### Cell culture and transfection
To avoid the unspecific activation of endogenous PIEZO1, we used throughout this study HEK293 cells KO for PIEZO1 (ref. [25]), named HEK-P1KO. These cells were a gift from Dr. Ardem Patapoutian (The Scripps Research Institute, La Jolla, CA, USA) and Dr. Eric Honoré (Institut de Pharmacologie Moléculaire et Cellulaire, CNRS, Valbonne, France) and were not authenticated. HEK-P1KO cells were maintained in Dubecco's Modified Eagle's Medium−high glucose (DMEM) supplemented with GlutaMax (Gibco, Life Technologies), 10% heat inactivated-fetal bovine serum (Gibco, Life Technologies), 100 units mL⁻¹ penicillin and 100 µg mL⁻¹ streptomycin (Gibco, Life Technologies) incubated at 37 °C in a 5% CO₂ atmosphere. Cells were passaged twice a week using 0.05% trypsin-EDTA (Gibco, Life Technologies) and used between passages 15−25 for the experiments. Cells were seeded in poly-L-lysine-treated 9-mm coverslips for patch-clamp at 5% confluence and poly-L-lysine-treated 12 mm coverslips for calcium imaging at 10−20% confluence 1 day before transfection.

Transfection was carried out using the calcium phosphate method in 35-mm well plates containing the seeded cells in the corresponding coverslips. We transfected 1 or 4 µg (for mP1 or Y2464C respectively) of plasmid per 35-mm dish for cell-attached experiments, 0.27 µg for whole-cell, and 2 µg for calcium imaging, and 5 µg per 100-mm dish for cell-surface biotinylation assay (unless stated otherwise).

### Molecular biology
mPIEZO1-IRES-eGFP was a gift from Dr. Ardem Patapoutian (Addgene plasmid #80925). Site-directed mutagenesis was carried out by PCR amplifications using the Q5® Hot Start High-Fidelity DNA Polymerase (New England BioLabs) using primers listed in Supplementary Data, followed by ligation with KLD Enzyme Mix (New England BioLabs).

For the mutant E2257K/E2258K/D2264K/Y2464C, we realized sequential site-directed mutagenesis, in which we first introduced E2257K/E2258K on the Y2464C template, and then introduced D2264K on the E2257K/E2258K/Y2464C background.

Chemo-competent bacteria (NEB 5-alpha Competent *E. Coli*, New England BioLabs) were transformed with the constructs and growth in LB medium containing 100 µg mL⁻¹ ampicillin.

In order to construct the mP1-IRES-GCaMP6 plasmid, we amplified the sequence corresponding to GCaMP6 in the rP4X4-GCaMP6 vector by PCR (ref. [46]), with the following primers: forward: 5′-AT AATATGGCCACAACCATGGGTTCTCATCATCATCATCATC-3′, reverse: 5′-AAACTTAAGCTTGGCCGGCCTCACTTCGCTGTCATCATTTG-3′, and amplified the full mP1-IRES-eGFP vector excluding eGFP with the

following primers: forward: 5′-GGCCGGCCAAGCTTAAGTTTAAAC-3′, reverse: 5′-CATGGTTGTGGCCATATTATCATCG-3′. Corresponding fragments were re-ligated with the Hifi DNA assembly master mix (New England BioLabs). For constructing the Hemagglutinin (HA) tagged Y2464C-mPIEZO1 (Y2464C PIEZO1-HA) plasmid, the HA tag was inserted at the C-terminus by PCR. We amplified the Y2464C-mPIEZO1-IRES-eGFP vector with the following primers (each one containing a part of the HA tag): forward: 5′-GCCGGATTATGCGTAGAAGCTTGG CGCGCCT-3′, reverse: 5′-ACATCATACGGATACTCCCTCTCACGTGTCC AC-3′, followed by treatment with KLD Enzyme Mix (New England BioLabs). All constructs were verified by DNA sequencing.

## Whole-cell patch-clamp electrophysiology

Whole-cell patch-clamp recordings were performed 24 h after transfection using an EPC10 USB amplifier (HEKA). Recordings were sampled between 5–20 kHz and filtered at 2.9 kHz. Traces were filtered offline at 1 kHz for analysis. Borosilicate pipettes of 4–6 MΩ were filled with an internal solution containing (in mM) 140 CsCl, 5 EGTA, 5 HEPES, 1 MgCl₂, 0.4 Na₂-GTP and 20 TEA (tetraethylammonium), pH 7.3 adjusted with CsOH. The external solution contained (in mM) 140 NaCl, 10 HEPES, 2 CaCl₂, 2 MgCl₂, and 10 glucose, pH 7.3 adjusted with NaOH. Osmolarity was adjusted to 290–300 mOsmol kg⁻¹ with glucose.

Cell poking was assessed with a Sensapex uMp micromanipulator (Sensapex, Finland) controlled with the Sensapex PC suite. Stimulations were performed at a velocity of 1 μm ms⁻¹ using only the z-axis of the manipulator. The interval between two adjacent stimulations was set to 10 s. Stimulations were made with a 3–4 μm blunt pipette placed at 45° related to the recording chamber.

DTT perfusion was carried out by gravity flow onto the recording chamber and an agar bridge with 3 M KCl was used to maintain the offset; Gd³⁺ and RR perfusion was carried out by gravity flow with a fast perfusion exchanger (SF-77B, Warner Instruments) externally controlled with the PATCHMASTER 2 × 91 program (HEKA).

## Cell-attached electrophysiology

48 h after transfection, cell-attached recordings were performed at room temperature with an external solution containing (in mM) 140 KCl, 10 HEPES, 1 MgCl₂, and 10 glucose, pH 7.3 adjusted with KOH. Pipettes of 2–3 MΩ were filled with the pipette solution containing (in mM) 130 NaCl, 5 KCl, 10 HEPES, 1 MgCl₂, 1 CaCl₂, and 10 TEA-Cl, pH 7.3 adjusted with NaOH. Osmolarity was adjusted to 290–300 mOsmol.kg⁻¹ with glucose. Recordings were acquired using an EPC10 USB amplifier (HEKA) sampled at 20 kHz and filtered externally for 50–60 Hz with a HumBug (Quest Scientific) and filtered offline at 1 kHz for analysis. Mechanically activated currents were achieved by using a high-speed pressure clamp directly attached to the patch holder (HSPC-2, ALA Scientific). To remove membrane tension due to patch-clamping, a pre-pulse of +5 mm Hg was applied for 5-s, as previously described[32], just before the challenging 300-ms pulse. Pressure-response protocols were carried out from 0 to −100 mm Hg at a holding potential of −80 mV. The +5 mm Hg pre-pulse did not elicit any current nor modify the baseline.

To resolve single-channel events in the cell-attached configuration, pipettes were coated with Sylgard 184 (Dow Corning Co.), fire polished to yield resistances of 8–11 MΩ, and filled with a solution containing (in mM) 130 NaCl, 5 KCl, 10 HEPES, 1 MgCl₂, 1 CaCl₂ and 10 TEA-Cl, pH 7.3 adjusted with NaOH. The bath solution contained (in mM) 140 KCl, 10 HEPES, 1 MgCl₂, and 10 glucose, pH 7.3 adjusted with KOH. Osmolarity was adjusted to 290–300 mOsmol kg⁻¹ with glucose. The holding potential was −80 mV. Data were acquired at a sampling rate of 20 kHz for pressure-gated currents or 40 kHz for light-gated currents, and low-passed filtered at 2.9 kHz. Data were re-filtered offline to 1 kHz and channel events were detected using TAC 3.0 software

(Bruxton Co.), and conductance levels were measured by all-points amplitude histograms fitted to Gaussian distributions.

## Light-gating stimulations

Cells were incubated for 20 min (unless stated otherwise) with 200 μM MAT at room temperature in an external solution containing (in mM) 140 NaCl, 10 HEPES, 2 CaCl₂, 2 MgCl₂, and 10 glucose, pH 7.3 adjusted with NaOH. Light pulses at 365- and 530-nm were delivered with a LED controller (DC4104, Thorlabs or Ultra-high-power, Prizmatix) with a power (unless stated otherwise) of 0.78–0.20 W cm⁻² (365 nm) and 0.5–0.12 W cm⁻² (530 nm) using the ×20 objective of the inverted microscope. External control was carried out with PATCH-MASTER 2 × 91.

Unless otherwise stated, all light stimulations in whole-cell configuration were carried out by a single irradiation at 365 nm for 200-ms, immediately followed by another irradiation at 530 nm for 800-ms. The same protocol was used for cell-attached patches, except that irradiations lasted 300 ms for 365 nm and 1 s for 530 nm.

To assess light-gated currents in different mP1 mutants, we first irradiated cells before testing cell poking stimulation. Only the cells that endured both protocols were used for analysis.

## Ion permeability

Ion permeability was determined in the whole-cell configuration. The control external solution contained (in mM) 150 NaCl, 10 HEPES and 10 glucose adjusted to pH 7.3 with NaOH. The internal solution contained (in mM) 150 CsCl, 10 HEPES adjusted to pH 7.3 with CsOH. We performed a set of 5 irradiations in the external control solution at different holding potentials from −80 mV to +80 mV, before changing the external solution to an isethionate-based solution or external Na⁺ reduced solutions. 1.5-min after solution exchange, we performed a second set of five irradiations at the same holding potentials. The isethionate-based external solution contained (in mM) 150 Na-isethionate, 10 HEPES, and 10 glucose adjusted to pH 7.3 with NaOH. Reduced external Na⁺-based solutions were as follows: (1) Na⁺₍out₎ 80: 80 mM NaCl, 120 mM mannitol, 10 mM HEPES, and 10 mM glucose; (2) Na⁺₍out₎ 40: 40 mM NaCl, 200 mM mannitol, 10 mM HEPES and 10 mM glucose, and (3) Na⁺₍out₎ 10: 10 mM NaCl, 280 mM mannitol, 10 mM HEPES and 10 mM glucose. All external Na⁺-based solutions were adjusted to pH 7.3 with NaOH. An agar bridge with 3 M KCl was used for all experiments and the junction potential was corrected before calculating reversal potentials $E_{rev}$. $E_{rev}$ values were determined by the difference between $E_{rev}$ measured in solution with reduced Na⁺₍out₎ content and $E_{rev}$ measured in the external control solution (expected to be close to 0 mV). $P_{Na}/P_{Cs}$ relative permeability was calculated adjusting $E_{rev}$ to the Goldmann-Hodgkin-Katz equation assuming no chloride permeability ($P_{Cl}/P_{Cs} = 0$), which simplifies to

$$E_{rev} = 1000 \times \frac{RT}{F} \ln\left( \frac{\frac{P_{Na}}{P_{Cs}} [Na]_{out}}{[Cs]_{in}} \right) \qquad (2)$$

where R is 8.314 J mol⁻¹ K⁻¹, T is 296.15 °K, F is 96,485 C mol⁻¹, $[Na]_{out}$ is the external Na⁺ concentration, which was varied from 151, 86, 41, to 14 mM, and $[Cs]_{in}$ is the internal Cs⁺ concentration, which was 154 mM. Because coefficient activities are the same between Na⁺ and Cs⁺, no conversion to ion activities was needed.

## Calcium imaging

After 20-min incubation with 200 μM MAT, seeded cells coverslips were mounted on the imaging chamber and perfused with 250 μL NPSS solution, containing (in mM) 140 NaCl, 5 KCl, 1 CaCl₂, 2 MgCl₂, 1 HEPES, 10 glucose, pH 7.32 adjusted with NaOH, 290–310 mOsmol kg⁻¹. Epifluorescence imaging experiments were performed with an inverted fluorescence microscope (IMIC2000 digital microscope, TILL Photonics) equipped with a polychrome-V (TILL Photonics), and a

camera (Hamamatsu orca flash 4.0 V2). The microscope was controlled by a computer running Live Acquisition Software 2.6.0.29 (TILL Photonics). The light source was from the 150 W Xenon lamp filtered at appropriate wavelength for eGFP by the optical filters mounted at the computer-controlled filter slider for GCaMP6 excitation at 491 nm for 100 ms, or for mOP1 irradiation at 365 nm or 530 nm for 1 s, subsequently passing the dichroic mirror and the emission filter. At $t_0$, 250 μL of NPSS, supplemented or not with 20 μM Yoda1 (for a final concentration of 10 μM), were added to the imaging chamber, followed immediately after by fluorescence acquisition at 510 nm using an Olympus ×40 NA 0.75 objective at a sampling rate of 0.2 Hz (1 image per 5 s). At $t_{19min}$, 500 μL of NPSS supplemented with 20 μM of Yoda1 (for a final concentration of 10 μM) or 10 μM of ionomycin (for a final concentration of 5 μM) with or without Yoda1 were added.

## Cell-surface biotinylation assay

Cell-surface biotinylation assay was carried out 24 h after transfection of HEK-P1KO cells with plasmid encoding Y2464C PIEZO1-HA. Cells were washed three times in ice-cold PBS+ solution (137 mM NaCl, 2.7 mM KCl, 10 mM $Na_2HPO_4$, 1.8 mM $KH_2PO_4$ supplemented with 1 mM $MgCl_2$, 0.4 mM $CaCl_2$ and adjusted to pH 8.0 at 4 °C) and incubated with 200 μM MAT for either 0 min, 20 min, or 30 min and washed three time in PBS+ to remove excess of MAT. Cells were then incubated in sNHS-SS-Biotin (Thermo Fisher) 2 mM in PBS+ for 30 min under gentle agitation and quenched with 20 mM Tris in PBS+. Cells were incubated for 60 min at 4 °C in lysis buffer containing: HEPES, 100 mM NaCl, 5 mM EDTA, 1% Triton-X, pierce protease inhibitor tablets (Thermo Fisher, Waltham, USA). Samples were centrifuged at 21,000 × g at 4 °C, and input collected at this stage. Following this, NeutrAvidin-agarose (Thermo Fisher) resin was added to lysis samples prior to a pre-cleaning step with lysis buffer. The resin was then washed three in washing buffer (20 mM Hepes, 500 mM NaCl, 5 mM EDTA, 1% Triton-X-100, pierce protease inhibitor tablets) and two times in lysis buffer. Samples were then resuspended in 70 mM DTT and NuPage LDS sample buffer (Thermo Fisher) and boiled for 10 min at 95 °C. The supernatant was loaded and run onto Mini-PROTEAN TGX 7.5% gels using TGS running buffer (BioRad). The proteins were then transferred onto nitrocellulose membrane using the TransBlot Turbo system (BioRad) and ran at 1.3 A or 2.5 A at 25 V for 10–20 min for one or two gels, respectively. Membranes were blocked for 1 h in PBST (137 mM NaCl, 2.7 mM KCl, 10 mM $Na_2HPO_4$, 1.8 mM $KH_2PO_4$ supplemented with 1% non-fat milk, 0.5% BSA and 0.05% Tween-20 and adjusted to pH 7.2 at 4 °C) and incubated with mouse monoclonal anti-HA primary antibody (Invitrogen 26183) diluted 1:500 in TBST overnight at 4 °C. Membranes were incubated with Goat anti-Mouse HRP secondary antibody (Invitrogen 31430) diluted 1:10,000 for 2 h at ambient temperature. Revelation was done by Amersham ECL select western blotting detection reagent (GE Life Sciences) and chemiluminescence was assessed using the Amersham Imager 600.

## Molecular modeling

To test if any residue on mP1 could match with I328 on the rP2X2, we performed a structural alignment between the PIEZO structure (PDB ID: 5Z10)[14] and the rP2X2 model[19]. Because there is low sequence identity between PIEZO1 and P2X2, we performed the alignment between TM1 (from G30 to E59) and TM2 (from Q321 to L353) of rP2X2 and TM37 (from K2185 to V2219) and TM38 (from L2461 to S2489) of the mP1, respectively, with Align function on PyMol 2.5.4.

To address the conformations adopted by MAT in the mOP1, we performed a series of molecular dockings using as a template the curved mP1 (PDB ID: 7WLT)[17], and the flattened mP1 (PDB ID: 7WLU)[17]. In-silico mutation was carried out during the optimization of side chain positions and was performed with the software Scwrl4[47], while keeping the main chain rigid. The structure of the protein and MAT was converted to pdbqt files with the software Open Babel 2.4.1. Covalent

docking was then performed with the software smina (based on AutoDock Vina 1.1.2)[48]. The box for docking has been defined around the mutated cysteine residue, with a size of 30 Å in each direction. Covalent docking forced one of the two carbons of the maleimide moiety (positions 3 and 4) of MAT to be at 1.6 Å from the sulfur atom of the introduced cysteine to allow reactivity.

## Statistics and reproducibility

MestReNova 14.2.3–29241 was used as NMR analysis software. Patch-clamp experiments were analyzed using Fitmaster 2 × 73 (HEKA), TAC 3.0, and TACFit 3.0 softwares (Bruxton Co.) or IgorPro 6.36 (Wavemetrics) and calcium imaging was analyzed with ImageJ 1.53a. Pressure-response curves in cell-attached were fitted with a Boltzmann equation of the form

$$I(P) = I_{min} + \frac{I_{max}}{\left(1 + \exp\left(\frac{(P_{50}-P)}{s}\right)\right)} \tag{3}$$

where $I$ is the peak of the stretch-activated current at a given pressure, $I_{min}$ is the minimum current, $P$ is the applied patch pressure (in mm Hg), $P_{50}$ is the pressure value that evoked a current value which is 50% of maximal current $I_{max}$, and $s$ reflects the current sensitivity to pressure. Whole-cell fading currents, $\tau_{inact\ at\ 365\ nm}$ and $\tau_{deact\ at\ 530\ nm}$, were fitted with a single exponential equation

$$I(t) = A \exp(-t/\tau) \tag{4}$$

where $t$ is the time, $\tau$ is the time constant at 365 nm or 530 nm, and $A$ is the current amplitude. Detection of single-channel events was carried out by using 50% of the single-channel current amplitude as the detection threshold. Current amplitudes were determined by fitting data to a sum of $n$ Gaussians, using maximum likelihood methods

$$f(x) = \sum_{i=1}^{n} \frac{a_i}{\sigma_i \sqrt{2\pi}} \exp\left[-\frac{(x-A_i)^2}{2\sigma_i^2}\right] \tag{5}$$

where $f(x)$ is the total probability density of a given amplitude value x, $A_i$ is the $i$th channel amplitude, $\sigma_i$ is the standard deviation of the $i$th channel amplitude, and $a_i$ is the fraction of the data represented by the $i$th amplitude. Conductance was determined by dividing current amplitude by the holding potential (−80 mV). Statistics and graphics were carried out with GraphPad Prism 9 or IgorPro 6.36. All samples were tested for normality by Shapiro–Wilk test and Grubbs test for outlier detection before selecting the appropriate tests for statistical sample differences. Unless otherwise stated, we used Mann–Whitney or Kruskal–Wallis test for sample comparison. No statistical method was used to predetermine sample size. For electrophysiological experiments, we excluded from the analyses cells that displayed excessive or unstable leak currents (e.g., >500 pA for whole-cell recordings). For patch-clamp DTT experiments, we excluded from the analyses cells that displayed unstable baseline currents due to irregular DTT perfusion. For single-channel recordings, we excluded patches that contained no conductance. For calcium imaging, we excluded cells that did not respond to the Yoda1 or ionomycin control or cells for which basal fluorescence (in the absence of Yoda1) was high. Randomization is not relevant to this study, as samples are not required to be allocated into experimental groups.

## Reporting summary

Further information on research design is available in the Nature Portfolio Reporting Summary linked to this article.

# Data availability

All data generated in this study are provided in the main text and Supplementary Information. Previously published structures from the

PDB can be accessed via accession codes: 5Z10, 7WLT, 7WLU. Source data are provided with this paper.

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

## Acknowledgements

We are grateful to Drs. A. Patapoutian for proving mPIEZO1-IRES-eGFP and E. Honoré for providing HEK-P1KO. We thank Dr. Kate Dunning for critical reading of the manuscript and R. Vauchelles and Dr. P. Carl for technical support for calcium imaging. This work has benefitted from support provided by the University of Strasbourg Institute for Advanced Study (USIAS) for a fellowship to F.A.P., within the French national program "Investment for the future" (Idex-Unistra) to T.G., by the Agence Nationale de la Recherche (Grant ANR-20-CE14-0016-02) to T.G., by the doctoral school ED414 (to B.A.), by the International Center for Frontier Research in Chemistry (Fondation Jean-Marie Lehn) (Labex CSC-TGR-18) to T.G., by The Région Grand Est to T.G., by the "École Universitaire de Recherche" Euridol (Programme d'investissement d'Avenir, ANR-17-EURE-0022) and was achieved within the NeuroStra Interdisciplinary Thematic Institute of the ITA 2021-2028 program of the University of Strasbourg, CNRS and Inserm. This work was also supported by the Interdisciplinary Thematic Institute SysChem via the Idex Unistra (ANR-10-IDEX-0002) within the program Investissement d'Avenir to T.G.

## Author contributions

F.A.P. performed cysteine mutagenesis, cell culture, patch-clamp electrophysiology, and analyzed data. M.B. and D.B. performed patch-clamp electrophysiology and analyzed data. A.M. performed calcium imaging, molecular biology, cell culture, patch-clamp electrophysiology, and analyzed data. F.C. performed molecular biology, and cell-surface biotinylation assay, T.C. performed calcium imaging, B.A. performed the chemical synthesis of MAT. A.T. performed molecular docking. T.G. analyzed data, conceived the research, supervised the study, obtained funding, and wrote the paper. All authors discussed the results and edited the manuscript at all stages.

## Competing interests

The authors declare no competing interests.
