## [Peer Review File · Nature Communications]

Reviewers' Comments:

Reviewer #1:

Remarks to the Author:

In this study, Peralta et al. utilize a covalently linked azobenzene-based photoswitch to enable light-induced activation of Piezo1 channels. Using electrophysiology, they show that optical and mechanical activation of Piezos display similar functional properties. They further demonstrate the applicability of this method to calcium imaging in cell-based assays. Finally, they attempt to interrogate the molecular mechanisms through which the photoswitch might promote channel activation.

Overall, this study is conceptually interesting and describes a potentially valuable tool for studying Piezo channels and is therefore appropriate for publication in Nature Communications. However, prior to acceptance of the manuscript, there are several points that need to be addressed.

Comments:

It is important to note that the results from the study referenced in line 57-59 regarding ASIC-Piezo1 chimeras were later shown to be irreproducible and were likely artifacts from endogenous Piezo1 in the cell line used in those experiments (Dubin et al., 2019, PMID: 28426961).

In lines 71-75, it might be nice to comment on the free energy differences between open and closed states of P2X receptors, as a point of comparison to Piezo since the same photoswitch is used to activate both.

The claim that MAT labelling abolishes currents in the G2463C mutant may also be explained by inherently low channel activity/trafficking. In extended data figure 2c, only one cell appears to show responses with lower amplitude than seen in other mutants.

It would be good to see calcium imaging responses in response to mechanical activation to be able to more easily compare responses in this system.

The data suggesting that there are three distinct conductance levels are not convincing. The patches appear to be noisy before and during light application. Does histogram analysis of events recorded from all patches reveal three distinct gaussian fits?

The channel openings in deltaBlade mOP1 could use some quantification (for example increase in current amplitude following light exposure)

It would be interesting to see if the E2257K/E2258K/D2264K mutants that slowed inactivation and deactivation kinetics in Lewis & Grandl 2020 (PMID: 31968259) in the context of mOP1 to see if they recapitulate what is seen in deltaBlade mOP1.

Due to the overall structural similarity between Piezo1 and Piezo2, would generating a mOP2 construct be straightforward? This would potentially be a very useful tool for the field as unlike Piezo1, there are no known non-mechanical activators of Piezo2.

Reviewer #2:

Remarks to the Author:

Peralta and colleagues design a light-gated Piezo1 channel based on their previous development of a light gated P2X receptor. A single point mutation at position Y2464C allows attachment of a photoswitch (MAT) with trimethyl ammonium as the channel-opening moiety. Overall, I find the study concise and elegant, yielding an interesting tool to the growing palette of azobenzene-based light gated channels/receptors. However, in my opinion the current study doesn't represent a major development in the field for three main reasons: 1- It is not conceptually different than previous reports (especially from the same group). In support, it did not require intensive screens. 2- the ligand is not really a specific Piezo1 ligand, rather a bulky structure to induce mechanical

force on the channel for opening. Perhaps this is why light-induced opening of channel is so negligible (~5% of total current). Perhaps another ligand (or chemical moiety) would have exhibited better performance. 3- No major insights have been gained with mOP1.

Major-

1. I find that the minute light-induced effects (5% of total current) to border impracticality. What can be done with 5% currents? I do not think this kind of tool may be useful in cells (in vitro or in vivo) or even for studying the biophysics of the channels (see comment #2, 3 below). Looking at previous reports using P2X receptors (which were the basis of this report), the authors obtained ~60% openings (Habermacher eLIFE, 2016). What do the authors make of this? And, what does 5% of I_{max} represent? Is this due to partial labeling of channels (e.g., labeling of only 5 % of channels on the membrane) or does it represent partial opening of channel? I think this should be better discussed and improved with another photoswitch. MAT does not seem to be specific for Piezo1, perhaps extant photoswitches with larger chemical moieties instead of trimethyl-ammonium?
2. The authors focus their attention of the description of the biophysical properties of mOP1. However, owing to the minute current, many of these data are likely shifted. Can the authors show the effects with relationship to current size (prior normalization etc.) compared to mP1?
3. Why is it that mOP1 can be constitutively potentiated by Yoda1, although mOP1 undergoes very rapid return to baseline in the presence of YODA1. This argues against the basic description of mOP1 that shows that they are "similar".
4. I find the Ca²⁺-imaging to be completely erroneous. The authors show that false Ca²⁺-transients are obtained when naïve cells are irradiated by 365 nm (extended 4a,b). In fact, these artifacts are on par with the supposedly real signals obtained in cells expressing mOP1 and treated with MAT. Moreover, the Ca²⁺-responses are so small, I wonder if they have any physiological effect? This experiment raises major concerns regarding the validity of the Ca²⁺ observations, and I therefore find the authors' claim : "These data, therefore, demonstrate that mOP1 can be used to simply interrogate PIEZO function in cell imaging systems" to be wrong. I would remove this whole section.
5. Cell attached experiments suggest ~25% labeling of channels, which is very different than the 5% observed in whole cell current. Why is this?
6. I do not understand how prolonged (30 min) incubation with MAT could induce the reductions in activity of the channels if it doesn't change their expression on membrane. Is MAT active in trans? This should be better described.

Reviewer #3:

Remarks to the Author:

This study engineers a novel bio-actuator coined mOP1 which endows photosensitivity to mechanosensitive PIEZO1 channel. The authors found that the Y2464C mutation of mouse PIEZO1 with maleimide ethylene azobenzene trimethyl ammonium (MAT) treatment could respond to 365 nm illumination. They further biophysically characterized the mOP1 and found that, compared with mechanical stimuli, it has a similar inactivation and recovery process but a very slow time constant, cation selectivity, voltage dependency and Yoda1 potentiation. They also showed mOP1 could be used in cell calcium imaging. Finally, they investigated the light-gating mechanism of mOP1. Under irradiation (365 nm), mOP1 shows a three conductance levels corresponding to three MAT molecules. They also identified that the detachment of the cap and the blade is necessary for light gating and the blades play an important role in rapid sensing MAT photoisomerization and exquisite inactivation and deactivation process.

This study designed a light-sensitive PIEZO1 for the first time, providing an excellent paradigm for the design of PIEZO1-based bio-actuator. The authors revealed the biophysical properties and light-gating mechanism of mOP1, which facilitates the further utilization and modification of mOP1. However, the light-gated current of mOP1 is small (tens of pA), and the time to reach the current peak and the inactivation time are significantly slower than those under mechanical force stimulation. Therefore, more evidence is needed to support whether the mOP1 system can mimic mechanical stimulation in physiological process. Meanwhile, their optical control system is an artificial tool, so the authors should focus on the application of the mOP1 instead of the light gating mechanism. In addition, the following comments need to be addressed.

- 1) As mentioned above, the activation and inactivation time of mOP1 under irradiation is significantly slower than that of mechanical stimulation, so I am curious that if mOP1 system can distinguish PIEZO1 mutations with different biophysical properties? Given the inactivation time of PIEZO1 significantly affects its physiological function (e.g. red blood cell volume), can mOP1 in this study mimic the function of PIEZO1 channels under physiological conditions through illumination?
- 2) Does the light-gated currents produced by mOP1 change with light intensity? Is it possible to increase the currents of mOP1 in this way?
- 3) Figure 1d shows that the inactivation of WT is similar to that of Y2464C, but in Extended Data Figure 4a, the fluorescence signal decrease of Y2464C under the Yoda1 is significantly faster than that of WT. How to explain this difference?
- 4) Does MAT covalent binding affect the distribution of Piezo channels on the cell membrane?
- 5) MAT-treated cells expressing mP1 also produce small light-gated responses in the presence of Yoda1 (Extended Data Figure 4). So how much of the increase in calcium response of mOP1 after application of Yoda1 is due to the effect of this non-specific signal? It is recommended to add a control of mOP1 without MAT treatment.
- 6) In line 231, the conclusion that the $23.6 \pm 7.6\%$ channels is labelled cannot be simply concluded from the comparison of the maximal currents under the two different stimulation modalities.
- 7) In the single-channel recording by irradiation, the conductance of o1, o2, and o3 are linearly proportional. Is it possible that o1, o2, and o3 correspond to 1, 2, and 3 channels, respectively, instead of the 3 states of one channel? How about the single channel conductance of Y2464C under mechanical stimulation? Furthermore, in figure 3e, it can be observed that mP1 also has flickering events that are smaller than full opening under pressure. Are these events different from o1 or o2 of mOP1?
- 8) Given that MAT can bind cysteine, does MAT affect the crosslinking of cysteine between the cap and the blade? Does DTT affect the binding of MAT to Y2464C? Besides illumination, it is better to show the currents under the mechanical stimulation in Figure 4b.
- 9) In line 302, Gd³⁺ is generally thought to affect MS channels by affecting cell membrane property, so it may be more reasonable to use a more recognized pore blocker (e.g. ruthenium red) for the Δ blade mutants.
- 10) The significant difference of inactivation time between light-gated current and mechanical-gated current suggests different gating mechanisms. What makes this difference?
- 11) The data of the Δ blade mutants can only prove that the blade can promote the mechanical transduction, but cannot prove that the blade can directly sense photoisomerization, and there may be a regulatory effect. Thus, the authors should compare the response with and without blade, and the subtitle in line 278 is less accurate.
- 12) The blades sense membrane perturbations caused by azobenzene photoisomerization to trigger channel opening. This seems to overestimate the necessity of blades in the light-gated motions, since Δ blade mOP1 is still able to respond to 365-nm light and it is unclear whether MAT interacts with phospholipids. Is it possible that the blades, as part of the Piezo channel, just assist in the opening of the central pore rather than directly sensing azobenzene photoisomerization?
- 13) "To detect spontaneous pore activity, we used Gd³⁺, a known mP1 pore blocker¹, and found specific reduction of the holding current upon Gd³⁺ application that was not observed in non-transfected cells (Figure 4e and Extended Data Figure 6)." The image and text do not match.

Other minor comment:

In line 304, the figure should be Extended Data Figure 8 instead of Extended Data Figure 6.

We are grateful to the reviewers for their constructive comments and suggestions on our manuscript. In response to their criticisms, we have carried out additional experiments and modified the manuscript incorporating new results. Altogether, we believe that these additional data further support our conclusions and considerably strengthen the manuscript.

A point-by-point response to the reviewers' comments is shown below. Our response is shown in blue, and quoted changes in the manuscript are highlighted in red.

Reviewer#1 (Remarks to the Author):

In this study, Peralta et al. utilize a covalently linked azobenzene-based photoswitch to enable light-induced activation of Piezo1 channels. Using electrophysiology, they show that optical and mechanical activation of Piezos display similar functional properties. They further demonstrate the applicability of this method to calcium imaging in cell-based assays. Finally, they attempt to interrogate the molecular mechanisms through which the photoswitch might promote channel activation.

Overall, this study is conceptually interesting and describes a potentially valuable tool for studying Piezo channels and is therefore appropriate for publication in Nature Communications.

We thank the reviewer for his/her positive comment on the manuscript.

However, prior to acceptance of the manuscript, there are several points that need to be addressed.

Comments:

It is important to note that the results from the study referenced in line 57-59 regarding ASIC-Piezo1 chimeras were later shown to be irreproducible and were likely artifacts from endogenous Piezo1 in the cell line used in those experiments (Dubin et al., 2019, PMID: 28426961).

We agree with this comment, and to avoid any confusion we have removed from the revised manuscript the sentence referring to the ASIC-PIEZO1 chimeras study.

In lines 71-75, it might be nice to comment on the free energy differences between open and closed states of P2X receptors, as a point of comparison to Piezo since the same photoswitch is used to activate both.

We thank the reviewer for this suggestion, and have added in the revised manuscript the value of the free energy difference between open and closed states of P2X receptor. These changes are now shown in the introduction (line 67) “However, a recent high-speed atomic force microscopy study has reported that it takes between 50 to 150 $k_B T$ (with k_B = Boltzmann constant, and T = temperature) to reversibly deform PIEZO1 structure from a curved, closed channel state into a flat, presumably open channel state²² (**Figure 1b**), a free energy difference (ΔG_o) that is not far from the free energy difference between open and closed states of P2X ($\sim 12 k_B T$)²³, and between *cis* and *trans* configurations of a single azobenzene molecule ($\Delta G_{cis-trans} \sim 20 k_B T$)²⁴, suggesting that azobenzene photoisomerization energy could, in principle, be partially converted into PIEZO channel gating (**Figure 1c**)”.

The claim that MAT labelling abolishes currents in the G2463C mutant may also be explained by inherently low channel activity/trafficking. In extended data figure 2c, only one cell appears to show responses with lower amplitude than seen in other mutants.

We thank the reviewer for this comment. As suggested, we have added a sentence to clarify this point. The text now reads as follows (line 123): “We also found evidence for **MAT** labeling at two other cysteine positions: V2467C and G2463C. Instead of producing light-gated currents, labeling at these positions either reduced inactivation rates induced by continued poking stimulations for V2467C, a feature also observed for Y2464C (**Supplementary Fig. 3f and g**), or abolished the few poking-evoked currents that were recorded in the absence of **MAT** labeling for G2463C (**Supplementary Fig. 3c and d**). This last result suggests that the G2463C mutation, on its own, may already alter channel activity and/or cell trafficking”.

It would be good to see calcium imaging responses in response to mechanical activation to be able to more easily compare responses in this system.

Although we agree with this comment, our cell imaging setup is not equipped with mechanical devices to allow mechanical activation. However, to satisfy the reviewer’s comment, **we have**

performed additional experiments in which light-induced calcium responses were compared to those induced by Yoda1, which is commonly used to specifically active PIEZO1 in cell imaging assays. Data are now included in Supplementary Fig. 5a and c of the revised version of the manuscript and the text was modified as follows (line 228): “When normalized to Yoda1, the specific PIEZO1 activator commonly used in cell imaging assays^{11,29,30}, these responses increased to $15.9 \pm 4.5\%$ ($n = 33$ cells), whereas no response was detected in the absence of MAT (Supplementary Fig. 5a and c)”.

Supplementary Fig. 5. GCaMP6 Ca^{2+} imaging with mOP1. **a**, Time-lapse of fluorescence variation (arbitrary units) depicting intracellular Ca^{2+} dynamic in response to 1 s irradiations at 365-nm in cells expressing Y2464C treated (thick black trace) or nor (dotted grey trace) with MAT. Note 1-s pre-irradiations at 530 nm just before the second and third irradiation at 365 nm. Yoda1 (10 μM) was applied as control (grey shadow). Corresponding fluorescent images are shown on the top at the indicated times. Scale bars, 10 μm . **b**, Time-lapse of fluorescence variation (arbitrary units) depicting intracellular Ca^{2+} dynamic in the continuous presence of 10 μM Yoda1 (arrow, applied just before the onset of the experiments) in response to 1 s irradiations at 365-nm in MAT-treated cells expressing either mOP1 (thick cyan trace) or mP1 (dotted cyan trace). Ionomycin (5 μM) was applied as control (grey shadow). Corresponding fluorescent images are shown on the top at the indicated times. Scale bars, 10 μm . **c**, Box plot of light-gated Ca^{2+} peak responses normalized to that of ionomycin or Yoda1 in function of irradiation cycles in the absence or presence of Yoda1. Responses are presented as median (center), 25-75 percentile (box) and 5-95 percentile (whisker). Outliers extending beyond whiskers ($1.25 \times$ whisker length) are shown. Indicated P -values are from Mann-Whitney test. Number of cells is indicated in parentheses. Note the presence of unspecific, light-gated responses only in the presence of Yoda1 and in the absence of MAT (-MAT, untreated cells) expressing either Y2464C or mP1.

The data suggesting that there are three distinct conductance levels are not convincing. The

patches appear to be noisy before and during light application. Does histogram analysis of events recorded from all patches reveal three distinct gaussian fits?

As requested by the reviewer, we have performed additional analysis of single-channel data to support the existence of three distinct conductance levels. We have analyzed the distribution of events from all cell-attached patches ($n = 15$), and have successfully detected 1,002 events with TACFit. Fitting the distribution with a sum of Gaussians (with the maximum likelihood methods) clearly indicated three distinct conductance levels.

Supplementary Fig. 6. Functional characterization of pressure- and light-evoked responses in the cell-attached configuration. **a**, Typical inward currents in the cell-attached configuration evoked by applying negative pressure through the recording pipette (-60 mm Hg for 300 ms) at a holding potential of -80 mV. Shown are two patches from cells that were transfected (upper trace) or not (lower trace) with mP1. To minimize resting membrane tension, a pre-pulse of +5 mm Hg was applied to the patch for 5 s prior to mechanical stimulations, as previously described¹. **b**, Current-pressure relationship of stretch-activated currents at -80 mV from patches expressing the indicated construct treated or not with MAT. Data were fitted with a Boltzmann equation giving I_{max} and P_{50} ($n = 11-13$ patches, indicated in parentheses). **c, d**, Corresponding I_{max} and P_{50} values. P -values are from Mann-Whitney test. **e**, Left, examples of single-channel currents at -80mV in the cell-attached configuration elicited by light irradiation (365 nm and 530 nm) from MAT-treated cells expressing mOP1. Channel openings are downward deflections. Right, sections of the recordings indicated by cyan lines below traces are shown at a

higher time resolution. Label c denotes closed channels and labels o_1 , o_2 and o_3 represent the three open conductance levels. Violet arrows indicate the start of irradiations. Corresponding all-point histograms with Gaussian fits are shown right of the traces. Data were acquired at 40 kHz and filtered at 1 kHz. **f**, Light-induced conductance (at -80 mV) of o_1 , o_2 and o_3 as a function of labeled subunits ($n = 15$ cell-attached patches). Each cell-attached patch is shown by a different symbol and color. Data were fitted with a linear regression (violet line) of the form $y = 7.5 \times x - 1.1$. Inset: cartoon depicting subunit labeling stoichiometry with one, two and three MAT molecules putatively corresponding, respectively, to o_1 , o_2 and o_3 . All data are presented as mean \pm s.e.m.. **g**, Distribution of detected events obtained from all cell-attached patches ($n = 15$) as a function of current amplitude. Amplitude histogram was fitted with three Gaussians and fitted values are shown on the top. Note the presence of three conductance level values that were very similar to those determined individually from each patch. **h**, Single-channel currents at -80 mV in the cell-attached configuration elicited by pressure from MAT-untreated cells expressing Y2464C. Channel openings are downward deflections. Labels c and o denote, respectively, closed and open conductance levels. Corresponding all-point histogram with Gaussian fits is shown right of the trace. Data were acquired at 20 kHz and filtered at 1 kHz.

Data are now shown in Supplementary Fig. 6g in the revised manuscript. Additional text and equation were inserted in the Methods. We have also modified the text as follows (line 285):

“These light-gated currents were nevertheless resolved in three distinct conductance levels of 6.8 ± 0.5 pS, 13.0 ± 0.6 pS and 21.8 ± 1.2 pS ($n = 15$ cell-attached patches analyzed separately), named, respectively, o_1 , o_2 and o_3 , and were linearly proportional with labeled subunits (**Figure 5f and Supplementary Fig. 6e and f**). Because the simultaneous presence of these distinct conductance levels was not systematically detected in every single patch, we further performed event analysis from all 15 cell-attached patches (1,002 events analyzed) and consistently revealed three distinct conductance levels that were very similar to those individually determined from each patch (**Supplementary Fig. 6g**).”.

To go further, we have also included single-channel recordings from mechanically-activated mP1-Y2464C (Supplementary Fig. 6h). Consistent with the fact that Y2464C inactivated more rapidly than the wild-type mP1, brief single-channel openings were recorded, but this was not analyzed further.

We hope these additional analysis and experiments resolve the issue.

The channel openings in deltaBlade mOP1 could use some quantification (for example increase in current amplitude following light exposure)

As requested, we have added some quantification of channel openings in Δ blade mOP1. We now present current density obtained at 365 nm for Δ blade mOP1 and the control Δ blade mP1

in Fig. 6f. For the remaining features, such as channel inactivation and deactivation, these cannot be quantified.

It would be interesting to see if the E2257K/E2258K/D2264K mutants that slowed inactivation and deactivation kinetics in Lewis & Grandl 2020 (PMID: 31968259) in the context of mOP1 to see if they recapitulate what is seen in deltaBlade mOP1.

We thank the reviewer for this excellent suggestion. **We have now added new experiments by introducing E2257K/E2258K/D2264K in mOP1 (named KKK mOP1). We found that the presence of the mutation increased τ_{inact} at 365 nm and τ_{deact} at 530 nm in a manner that was similar to poking stimulations.** Together these data demonstrate that light-induced azobenzene isomerization fairly mimics mechanical stimulations and further support the idea that destabilizing the interactions between the cap and blades impedes channel inactivation, as suggested by data obtained with in Δ Blade mOP1.

Figure 3. mOP1 recapitulates functional properties of selected PIEZO1 mutants. a, Whole-cell current inactivation at -80 mV elicited by 2-s activation at 365 nm (top) or 200-ms by cell poking (middle) and current deactivation (bottom) elicited by 530-nm light following 50 ms activation at 365 nm in cells expressing either mOP1, R2482H mOP1, or E2257K/E2258K/D2264K mOP1 (KKK mOP1). Fading currents were fitted with an exponential equation giving τ values. **b**, Average τ , time-to-peak and current amplitude values for 365-nm light (top), cell poking (middle) and 530-nm light (bottom) conditions (n = 13-25 cells) for mOP1 (left), R2482H mOP1 (middle) and KKK mOP1 (right). Comparison to control mOP1 with Mann-Whitney test or unpaired t test (for time-to-peak_{365 nm} and τ_{deact} 530 nm). *P*-values: ****< 0.0001, ***= 0.0007, **= 0.0049 for τ_{inact} 365 nm and $I_{365 nm}$, **= 0.0051 for τ_{inact} poking, *= 0.0196 and NS = 0.3538, 0.0932, 0.3361 and 0.2340 (from left to right). NS, not significant. All data are presented as mean \pm s.e.m..

These data are now shown in a new Figure 3, and a short paragraph was added entitled (line 194):

“mOP1 recapitulates features of selected PIEZO1 mutations

To further validate mOP1 application, we next sought to establish whether mOP1 can distinguish PIEZO1 mutations that were previously shown to alter channel biophysical properties. We focused on the single R2482H and triple E2257K/E2258K/D2264K mutations, which were previously described to slower mechanical-evoked inactivation and deactivation^{26,31}, and asked whether light can reproduce these mechanical-induced features. We therefore introduced these mutations into the Y2464C background (R2482H mOP1 and E2257K/E2258K/D2264K mOP1, named hereafter KKK mOP1) and labeled mutant-expressing cells with **MAT**. We found that light significantly increased both τ_{inact} at 365 nm and τ_{deact} at 530 nm in the two mutants, when compared to mOP1, in a manner that was similar to poking stimulations and published data^{26,31} (**Figure 3a and b**). These results therefore suggest that light-induced azobenzene isomerization recapitulates inactivation and deactivation kinetics induced by mechanical stimulations in PIEZO1 mutants.

We also found that light-gated currents significantly developed more slowly for both mutants, whereas no change was observed for mechanical stimulations (although KKK mOP1 currents tended to develop more slowly upon mechanical activation, $P = 0.0932$, **Figure 3b**). In addition, the presence of both mutations significantly increased light-gated current amplitudes, as compared to poking-evoked current amplitudes which remained unchanged (**Figure 3b**). As a consequence, light-gated currents represented ~40% of maximal poking-induced currents in both mutants ($38 \pm 6\%$ for R2482H mOP1 and $43 \pm 5\%$ for KKK mOP1), indicating that slowing inactivation kinetics increases light-gated current amplitudes.”

Due to the overall structural similarity between Piezo1 and Piezo2, would generating a mOP2 construct be straightforward? This would potentially be a very useful tool for the field as unlike Piezo1, there are no known non-mechanical activators of Piezo2.

We thank the reviewer for this suggestion, and actually, we are working on this issue. However, we feel that extension of the present study to PIEZO2 would go far beyond its scope, mainly because the focus of our study was limited to PIEZO1. We hope the reviewer understands.

Reviewer #2 (Remarks to the Author):

Peralta and colleagues design a light-gated Piezo1 channel based on their previous development of a light gated P2X receptor. A single point mutation at position Y2464C allows attachment of a photoswitch (MAT) with trimethyl ammonium as the channel-opening moiety. Overall, I find the study concise and elegant, yielding an interesting tool to the growing palette of azobenzene-based light gated channels/receptors.

We thank the reviewer for his/her positive comment.

However, in my opinion the current study doesn't represent a major development in the field for three main reasons: 1- It is not conceptually different than previous reports (especially from the same group). In support, it did not require intensive screens. 2- the ligand is not really a specific Piezo1 ligand, rather a bulky structure to induce mechanical force on the channel for opening. Perhaps this is why light-induced opening of channel is so negligible (~5% of total current). Perhaps another ligand (or chemical moiety) would have exhibited better performance. 3- No major insights have been gained with mOPI.

We understand the issue raised by the reviewer, and we take this opportunity to clarify our results: 1- The concept is completely different from what was previously reported (especially from our own group), because here we use a photoswitch to induce channel opening of a mechanically-activated ion channel instead of a ligand-gated ion channel. Despite the fact that the same photoswitch was used, we also show that the light-gating mechanisms are different between mechanically-activated and ligand-gated ion channels, further supporting for a new concept. The fact that screening was not intense is not indicative of a lack of novelty. 2- It is possible that other photoswitches with different chemical properties would exhibit better performance, but testing such compounds would require extensive chemical synthesis which would go way beyond the scope of this study. **However, to respond to the reviewer's comment, we have carried out additional experiments and now provide evidence that light-induced opening in gain-of-function mOPI mutants represents ~40% of total current.** 3- We believe that our tool can provide new insights into the biology of PIEZO1. In particular, we found that the blades contribute to channel inactivation, a result that is extremely difficult to assess with truncated channels that do not respond to mechanical stimulations.

Major-

1. I find that the minute light-induced effects (5% of total current) to border impracticality.

What can be done with 5% currents? I do not think this kind of tool may be useful in cells (in vitro or in vivo) or even for studying the biophysics of the channels (see comment #2, 3 below). Looking at previous reports using P2X receptors (which were the basis of this report), the authors obtained ~60% openings (Habermacher eLIFE, 2016). What do the authors make of this? And, what does 5% of I_{max} represent? Is this due to partial labeling of channels (e.g., labeling of only 5 % of channels on the membrane) or does it represent partial opening of channel? I think this should be better discussed and improved with another photoswitch. MAT does not seem to be specific for Piezo1, perhaps extant photoswitches with larger chemical moieties instead of trimethyl-ammonium?

We thank the reviewer for pointing this issue, and we take this opportunity to clarify the results. First of all, one cannot compare present data with those obtained previously by Habermacher eLIFE 2016, because and as a matter of fact, two different photoswitches have been employed (MAT in the present study and MAM in the previous report). Whereas MAT reacts with one single cysteine, MAM crosslinks between two, which results in a different gating mechanism (tweezers-like). Because of this difference, direct comparison cannot be made.

Second, to better show what 5% represents in terms of currents (prior to normalization), we now show and compare currents induced by light to those evoked by cell poking. Although non-normalized light-gated currents still represented $7.3 \pm 1.8\%$ of those induced by poking, they are not negligible as they represent 90 ± 23 pA (mean \pm s.e.m. [from 31 to 344 pA] for light versus $1,230 \pm 284$ pA [from 39 to 3,840 pA] for poking). We now show these values in new Supplementary Fig. 3e and have changed text as follows (line 111): “**On average, light-gated currents represented $5.3 \pm 0.7\%$ (mean \pm s.e.m.) of maximal current density evoked by the largest poking stimulations (or $7.3 \pm 1.8\%$ of maximal current amplitude, **Supplementary Fig. 3e)**”.**

Third, we now provide new data in which we demonstrate that light-gated currents increased to more than ~300 pA (that is ~40% of currents induced by cell poking) in mOP1 incorporating either the single gain-of-function R2482H mutation or the triple E2257K/E2258K/D2264K mutation. These data thus suggest potential use *in vivo*, in particular in mice carrying the gain-of-function R2482H.

Figure 3. mOP1 recapitulates functional properties of selected PIEZO1 mutants. **a**, Whole-cell current inactivation at -80 mV elicited by 2-s activation at 365 nm (top) or 200-ms by cell poking (middle) and current deactivation (bottom) elicited by 530-nm light following 50 ms activation at 365 nm in cells expressing either mOP1, R2482H mOP1, or E2257K/E2258K/D2264K mOP1 (KKK mOP1). Fading currents were fitted with an exponential equation giving τ values. **b**, Average τ , time-to-peak and current amplitude values for 365-nm light (top), cell poking (middle) and 530-nm light (bottom) conditions ($n = 13$ -25 cells) for mOP1 (left), R2482H mOP1 (middle) and KKK mOP1 (right). Comparison to control mOP1 with Mann-Whitney test or unpaired t test (for time-to-peak_{365nm} and τ_{deact} 530 nm). P -values: **** < 0.0001 , *** $= 0.0007$, ** $= 0.0049$ for τ_{inact} 365 nm and I_{365nm} , ** $= 0.0051$ for τ_{inact} poking, * $= 0.0196$ and NS $= 0.3538, 0.0932, 0.3361$ and 0.2340 (from left to right). NS, not significant. All data are presented as mean \pm s.e.m..

These data are now shown in a new Figure 3, and a short paragraph was added entitled (line 194):

“mOP1 recapitulates features of selected PIEZO1 mutations

To further validate mOP1 application, we next sought to establish whether mOP1 can distinguish PIEZO1 mutations that were previously shown to alter channel biophysical properties. We focused on the single R2482H and triple E2257K/E2258K/D2264K mutations, which were previously described to slower mechanical-evoked inactivation and deactivation^{26,31}, and asked whether light can reproduce these mechanical-induced features. We therefore introduced these mutations into the Y2464C background (R2482H mOP1 and E2257K/E2258K/D2264K mOP1, named hereafter KKK mOP1) and labeled mutant-expressing cells with MAT. We found that light significantly increased both τ_{inact} at 365 nm and τ_{deact} at 530 nm in the two mutants, when compared to mOP1, in a manner that was similar to poking stimulations and published data^{26,31} (Figure 3a and b). These results therefore suggest that light-induced azobenzene isomerization recapitulates inactivation and deactivation kinetics induced by mechanical stimulations in PIEZO1 mutants.

We also found that light-gated currents significantly developed more slowly for both mutants, whereas no change was observed for mechanical stimulations (although KKK mOP1 currents tended to develop more slowly upon mechanical activation, $P = 0.0932$, **Figure 3b**). In addition, the presence of both mutations significantly increased light-gated current amplitudes, as compared to poking-evoked current amplitudes which remained unchanged (**Figure 3b**). As a consequence, light-gated currents represented $\sim 40\%$ of maximal poking-induced currents in both mutants ($38 \pm 6\%$ for R2482H mOP1 and $43 \pm 5\%$ for KKK mOP1), indicating that slowing inactivation kinetics increases light-gated current amplitudes.”

We have also added a paragraph in the Discussion section as follows (line 380): “In addition, we found that this yield can be increased up to $\sim 40\%$ of mechanically-activated whole-cell currents in the gain-of-function R2482H, thus suggesting that light-gated activation might be sufficient to trigger, at least partially, physiological responses in mice carrying this mutation^{31,39}”

Last, we also provide new evidence that changing light intensity can tune current magnitude, suggesting irradiation systems that are more powerful than our LED can increase the amplitude of light-gated currents (see Supplementary Fig. 4a-c).

Supplementary Fig. 4. Functional characterization of light-induced inactivation and recovery of mOP1. a, Typical whole-cell currents at -80 mV recorded from the same cell expressing MAT-labeled Y2464C mutant (mOP1) in response to three successive cycles of 365-nm (violet) and 530-nm (green) at the indicated light intensity of the 365-nm LED. Inactivation currents were fitted with an exponential equation giving τ_{inact} values at

365 nm. **b**, Superimposed traces recorded from another cell of the second and third light-gated currents time-locked to light activation (arrow head). **c**, Average τ_{inact} (left), time-to-peak (middle) and current-density (right) values ($n = 7$ cells). Indicated P -values are from paired t -test for time-to-peak and τ_{inact} data, and from Wilcoxon matched-pairs signed rank test for current amplitude data. **d**, Typical traces of whole-cell inward currents evoked by two cycles (I_1 and I_2) of irradiation at 365-nm (200 ms) and 530-nm light (800 ms) separated by different time intervals of 5, 10, 20, 40 and 60 s. Each trace was recorded from a different cell. **e**, Average ratio (I_2/I_1) of currents induced at 365 nm in function of time interval ($n = 11-23$ cells). Indicated P -values are from Wilcoxon test. **f**, Typical traces of whole-cell inward currents evoked by two cycles of irradiation at 365-nm (200 or 50 ms) and 530-nm light (800 or 950 ms) separated by a time interval of 10 s. Note the absence of current inactivation during the 50-ms irradiation at 365 nm. **g**, Average ratio (I_2/I_1) of currents in function of irradiation time at 365 nm ($n = 11-15$ cells). Mann-Whitney test, P -value ***= 0.0004. All data are presented as mean \pm s.e.m..

We hope the Reviewer is now convinced by these clarifications and additional results.

2. The authors focus their attention of the description of the biophysical properties of mOP1. However, owing to the minute current, many of these data are likely shifted. Can the authors show the effects with relationship to current size (prior normalization etc.) compared to mP1?

As suggested by the reviewer (see the response to the comment above), we now show current size (prior normalization) of mOP1 and comparison with those induced by cell poking in mP1. These data are shown in Supplementary Fig. 3e in the revised manuscript.

3. Why is it that mOP1 can be constitutively potentiated by Yoda1, although mOP1 undergoes very rapid return to baseline in the presence of YODA1. This argues against the basic description of mOP1 that shows that they are “similar”.

We thank the reviewer for pointing out this issue for which we take the opportunity to clarify our initial statement. First of all, we recognize that our initial manuscript was not sufficiently clear to understand that our calcium imaging assay does not contain washing steps. In these conditions, Yoda1 cannot be washed out, and this therefore explains why mOP1 was constitutively potentiated by Yoda1. Second, it is true that the fluorescence signal decrease of Y2464C under Yoda1 was faster than that of wild-type mP1 (this point was also raised by reviewer #3, see point 3). This difference can actually be explained by a faster inactivation rate of the Y2464C mutant itself (not labelled by MAT) as compared to that of mP1 (see Supplementary Fig. 3f and g). If one assumes that labelling yield is low (which is likely to be the case here), then most of Yoda1-induced calcium responses in our system came from unlabeled channels. However, as shown in Supplementary Fig. 3f and g the difference in inactivation rates between the wild-type and Y2464C becomes very small, although significant,

when Y2464C was labeled with MAT. Therefore, our data show that MAT labeling tends to normalize inactivation rates, which further supports the similarity of channel inactivation between mOP1 and mP1.

We have therefore modified the text in the revised manuscript as follows (line 233): “**This plateau was expected because of the continuous bath presence of Yoda1 that constitutively gated PIEZO activity (there were no washing steps allowing Yoda1 removal for all experiments). In line with the fact that the MAT-unlabeled Y2464C mutant inactivated more rapidly than mP1 (Supplementary Fig. 3f and g), the decrease of fluorescent signal was also more rapid for mOP1 than for mP1. These data thus suggest that most of Yoda1-induced Ca²⁺ responses come from unlabeled channels**”.

We hope this issue is resolved.

4. I find the Ca²⁺-imaging to be completely erroneous. The authors show that false Ca²⁺-transients are obtained when naïve cells are irradiated by 365 nm (extended 4a,b). In fact, these artifacts are on par with the supposedly real signals obtained in cells expressing mOP1 and treated with MAT. Moreover, the Ca²⁺-responses are so small, I wonder if they have any physiological effect? This experiment raises major concerns regarding the validity of the Ca²⁺ observations, and I therefore find the authors' claim :”These data, therefore, demonstrate that mOP1 can be used to simply interrogate PIEZO function in cell imaging systems” to be wrong. I would remove this whole section.

First of all, it has to be stressed that these artifacts only occurred in the presence of Yoda1, not in its absence, as demonstrated by controls carried out in the absence of Yoda1 (mP1 treated with MAT) which revealed no signal in response to light (see Fig. 4c). However, as also suggested by reviewer #3 (see comment 5), **we have added new control experiments, in which light-gated calcium signals were monitored in Y2464C without MAT treatment both in the absence and presence of Yoda1 (now shown in Supplementary Fig. 5a and c of the revised manuscript). Clearly no signal was detected in the absence of Yoda1 confirming the high specificity of mOP1.** In the presence of Yoda1, and as already described in the initial version, we observed these artifactual signals in the absence of MAT treatment (both in mP1 and Y2464C) and only in the presence of Yoda1. Therefore, these controls demonstrate that these artifactual signals were independent of MAT and only occurred in the presence of Yoda1. We also show that mOP1 signals in presence of Yoda1 were significantly higher than controls

performed without MAT. How irradiation at 365 nm induced Yoda1-dependent calcium responses remains, however, unexplained.

Supplementary Fig. 5. GCaMP6 Ca^{2+} imaging with mOP1. **a**, Time-lapse of fluorescence variation (arbitrary units) depicting intracellular Ca^{2+} dynamic in response to 1 s irradiations at 365-nm in cells expressing Y2464C treated (thick black trace) or nor (dotted grey trace) with MAT. Note 1-s pre-irradiations at 530 nm just before the second and third irradiation at 365 nm. Yoda1 (10 μM) was applied as control (grey shadow). Corresponding fluorescent images are shown on the top at the indicated times. Scale bars, 10 μm . **b**, Time-lapse of fluorescence variation (arbitrary units) depicting intracellular Ca^{2+} dynamic in the continuous presence of 10 μM Yoda1 (arrow, applied just before the onset of the experiments) in response to 1 s irradiations at 365-nm in MAT-treated cells expressing either mOP1 (thick cyan trace) or mP1 (dotted cyan trace). Ionomycin (5 μM) was applied as control (grey shadow). Corresponding fluorescent images are shown on the top at the indicated times. Scale bars, 10 μm . **c**, Box plot of light-gated Ca^{2+} peak responses normalized to that of ionomycin or Yoda1 in function of irradiation cycles in the absence or presence of Yoda1. Responses are presented as median (center), 25-75 percentile (box) and 5-95 percentile (whisker). Outliers extending beyond whiskers (1.25 \times whisker length) are shown. Indicated P -values are from Mann-Whitney test. Number of cells is indicated in parentheses. Note the presence of unspecific, light-gated responses only in the presence of Yoda1 and in the absence of MAT (-MAT, untreated cells) expressing either Y2464C or mP1.

We have thus modified the text as follows (line 242): “Surprisingly, significant light-gated responses were also detected in MAT-treated cells expressing mP1 ($3.1 \pm 0.5\%$ of ionomycin signal, $n = 108$ cells); however, these signals appeared to be independent of MAT as they were observed in the absence of labeling in both Y2464C and mP1, and only in the presence of Yoda1 (Supplementary Fig. 5c)”.

We would like to keep the Yoda1 section in the revised form of the manuscript because it documents on the limits of the use of Yoda1 when irradiated at 365 nm. We hope these additional control experiments resolve this issue.

5. Cell attached experiments suggest ~25% labeling of channels, which is very different than the 5% observed in whole cell current. Why is this?

The reason for this discrepancy is currently not clear, but we can make a tentative interpretation. Recent work has suggested two gating models to activate PIEZO channels: the force-from-lipids and the force-from-filaments models. The relative contribution of these gating models to PIEZO1 activation is unclear in HEK cells, but it might explain the differences of the ratio of light-gated currents to mechanical-gated currents observed in whole cell versus cell-attached patches.

We have added few sentences dealing with this issue in the Discussion as follows (line 370): “First, whole-cell light-gated currents represent only a fraction (between 5 to 10%) of those mechanically activated by cell poking and it is unknown if this amount is sufficient to trigger physiological responses. However, we found that this amount increased to ~25% in cell-attached patches when currents were normalized to those induced by pressure. Therefore, the actual yield of light-gated activation is difficult to assess with high precision and seems to depend on the mechanical stimulation mode. In support of this hypothesis, it has been found that PIEZO1 can mediate both localized and whole-cell mechanical responses *via* two different gating models, the force-from-lipids and the force-from-filaments models³⁶⁻³⁸. Although the relative contribution of these gating models to PIEZO1 activation is unclear in HEK cells, they might explain the difference in the light-gated yield values”.

6. I do not understand how prolonged (30 min) incubation with MAT could induce the reductions in activity of the channels if it doesn't change their expression on membrane. Is MAT active in trans? This should be better described.

This point was also raised by reviewer #3 (see point 4). We have thus performed additional experiments to investigate whether or not MAT labeling induces any changes of PIEZO1 membrane expression. By determining cell-surface expression of biotinylated Y2464C PIEZO1-HA labelled with MAT for 20 or 30 min, we clearly provide evidence that labeling at

30 min does not change plasma membrane expression, suggesting that long incubation times >20 min might induce a defect on channel gating rather than channel internalization.

Supplementary Fig. 7. Cell-surface protein biotinylation assay of Y2464C PIEZO1-HA. **a**, Uncropped Western blot of biotinylated membrane-associated proteins (cell-surface) or total cell lysate (input) separated by SDS-PAGE from HEK-P1KO cells transiently transfected with 5, 7.5 or 10 µg of plasmid encoding hemagglutinin (HA) tagged Y2464C mPIEZO1 (Y2464C PIEZO1-HA). The blot was probed with anti-HA antibody. Note the presence of specific bands running at the expected molecular masses (~300 kDa), as previously described^{2,3}. Below is shown the same blot re-probed with anti-β-Actin. **b**, Uncropped Western blot of biotinylated membrane-associated proteins (cell-surface) or total cell lysate (input) separated by SDS-PAGE from HEK-P1KO cells transiently transfected with 5 µg of plasmid encoding Y2464C PIEZO1-HA. Before biotinylation, transfected cells were incubated with 200 µM MAT for 20 or 30 min (n = 2 independent replicates). Blots were probed with anti-HA antibody. Below input is shown the same blot re-probed with anti-β-Actin. Apparent molecular weights are indicated on the left of Western blots. NT, non-transfected.

The mechanism by which this reduction occurs is currently unknown, but as we show that MAT is not active in *trans* (no light-induced currents), it is possible that such reduction might be a consequence of the chemical modification of the Y2464C side chain, rather than active *trans*-MAT. Regardless the mechanism, we provide evidence that no reduction of PIEZO1 activity occurred at 20 min incubation, which was the time incubation used throughout the manuscript.

We have now added these data in a new Supplementary Fig. 7 and modified the text as follows (line 274): “This reduction of activity was not related to a change of PIEZO1 plasma membrane expression, as assessed by cell-surface protein biotinylation (**Supplementary Fig. 7**), suggesting that long incubation times >20 min might induce a reduction of PIEZO1 gating through an unknown mechanism”. We also included the experimental procedure in Methods.

Overall, we hope the reviewer appreciates the changes that were made to the manuscript and considers the issues as being resolved.

Reviewer #3 (Remarks to the Author):

This study engineers a novel bio-actuator coined mOP1 which endows photosensitivity to mechanosensitive PIEZO1 channel. The authors found that the Y2464C mutation of mouse PIEZO1 with maleimide ethylene azobenzene trimethyl ammonium (MAT) treatment could respond to 365 nm illumination. They further biophysically characterized the mOP1 and found that, compared with mechanical stimuli, it has a similar inactivation and recovery process but a very slow time constant, cation selectivity, voltage dependency and Yoda1 potentiation. They also showed mOP1 could be used in cell calcium imaging. Finally, they investigated the light-gating mechanism of mOP1. Under irradiation (365 nm), mOP1 shows a three conductance levels corresponding to three MAT molecules. They also identified that the detachment of the cap and the blade is necessary for light gating and the blades play an important role in rapid sensing MAT photoisomerization and exquisite inactivation and deactivation process.

This study designed a light-sensitive PIEZO1 for the first time, providing an excellent paradigm for the design of PIEZO1-based bio-actuator. The authors revealed the biophysical properties and light-gating mechanism of mOP1, which facilitates the further utilization and modification of mOP1. However, the light-gated current of mOP1 is small (tens of pA), and the time to reach the current peak and the inactivation time are significantly slower than those under mechanical force stimulation. Therefore, more evidence is needed to support whether the mOP1 system can mimic mechanical stimulation in physiological process. Meanwhile, their optical control system is an artificial tool, so the authors should focus on the application of the mOP1 instead of the light gating mechanism.

We thank the reviewer for his/her useful comments. As suggested, we have performed additional experiments with PIEZO1 mutants, further reinforcing the hypothesis that mOP1 fairly mimics mechanical stimulations. We have also modified the text to highlight potential applications of mOP1 and have included a paragraph in the discussion dealing with limitations of our tool.

In addition, the following comments need to be addressed.

1) As mentioned above, the activation and inactivation time of mOP1 under irradiation is significantly slower than that of mechanical stimulation, so I am curious that if mOP1 system can distinguish PIEZO1 mutations with different biophysical properties? Given the inactivation time of PIEZO1 significantly affects its physiological function (e.g. red blood cell volume), can mOP1 in this study mimic the function of PIEZO1 channels under physiological conditions through illumination?

We thank the reviewer for this excellent suggestion. We have thus carried out additional experiments by introducing the mouse R2482H gain-of-function mutation into mOP1, for which the equivalent mutation in human (R2456H) causes hereditary xerocytosis. We now provide evidence that light-induced azobenzene isomerization recapitulates inactivation kinetics induced by mechanical stimulations in this gain-of-function mutant. We believe that these additional experiments further validate the mOP1 system.

Figure 3. mOP1 recapitulates functional properties of selected PIEZO1 mutants. **a**, Whole-cell current inactivation at -80 mV elicited by 2-s activation at 365 nm (top) or 200-ms by cell poking (middle) and current deactivation (bottom) elicited by 530-nm light following 50 ms activation at 365 nm in cells expressing either mOP1, R2482H mOP1, or E2257K/E2258K/D2264K mOP1 (KKK mOP1). Fading currents were fitted with an exponential equation giving τ values. **b**, Average τ , time-to-peak and current amplitude values for 365-nm light (top), cell poking (middle) and 530-nm light (bottom) conditions ($n = 13-25$ cells) for mOP1 (left), R2482H mOP1 (middle) and KKK mOP1 (right). Comparison to control mOP1 with Mann-Whitney test or unpaired t test (for time-to-peak_{365nm} and $\tau_{deact\ 530\text{nm}}$). P -values: **** < 0.0001 , *** $= 0.0007$, ** $= 0.0049$ for $\tau_{inact\ 365\text{nm}}$ and $I_{365\text{nm}}$, ** $= 0.0051$ for $\tau_{inact\ poking}$, * $= 0.0196$ and NS $= 0.3538, 0.0932, 0.3361$ and 0.2340 (from left to right). NS, not significant. All data are presented as mean \pm s.e.m..

These data are now shown in a new Figure 3, and a short paragraph was added entitled (line 194):

“mOP1 recapitulates features of selected PIEZO1 mutations

To further validate mOP1 application, we next sought to establish whether mOP1 can distinguish PIEZO1 mutations that were previously shown to alter channel biophysical properties. We focused on the single R2482H and triple E2257K/E2258K/D2264K mutations, which were previously described to slower mechanical-evoked inactivation and deactivation^{26,31}, and asked whether light can reproduce these mechanical-induced features. We therefore introduced these mutations into the Y2464C background (R2482H mOP1 and E2257K/E2258K/D2264K mOP1, named hereafter KKK mOP1) and labeled mutant-expressing cells with **MAT**. We found that light significantly increased both τ_{inact} at 365 nm and τ_{deact} at 530 nm in the two mutants, when compared to mOP1, in a manner that was similar to poking stimulations and published data^{26,31} (**Figure 3a and b**). These results therefore suggest that light-induced azobenzene isomerization recapitulates inactivation and deactivation kinetics induced by mechanical stimulations in PIEZO1 mutants.

We also found that light-gated currents significantly developed more slowly for both mutants, whereas no change was observed for mechanical stimulations (although KKK mOP1 currents tended to develop more slowly upon mechanical activation, $P = 0.0932$, **Figure 3b**). In addition, the presence of both mutations significantly increased light-gated current amplitudes, as compared to poking-evoked current amplitudes which remained unchanged (**Figure 3b**). As a consequence, light-gated currents represented ~40% of maximal poking-induced currents in both mutants ($38 \pm 6\%$ for R2482H mOP1 and $43 \pm 5\%$ for KKK mOP1), indicating that slowing inactivation kinetics increases light-gated current amplitudes.”

These data therefore suggest potential use *in vivo*, in particular in mice carrying the gain-of-function R2482H.

2) *Does the light-gated currents produced by mOP1 change with light intensity? Is it possible to increase the currents of mOP1 in this way?*

We thank the reviewer for pointing out this issue. Actually, the light intensity of our LED system was set to the maximum. However, to investigate whether light intensity can change light-gated currents, **we have carried out additional experiments in which light intensity**

was decreased around 10-fold. We now show that both activation and inactivation kinetics, as well as current amplitude, were significantly decreased at this low intensity, clearly suggesting that mOP1 kinetics can be accelerated by more powerful devices. We provide these data in Supplementary Fig. 4a-c in the revised version of the manuscript.

Supplementary Fig. 4. Functional characterization of light-induced inactivation and recovery of mOP1. **a**, Typical whole-cell currents at -80 mV recorded from the same cell expressing MAT-labeled Y2464C mutant (mOP1) in response to three successive cycles of 365-nm (violet) and 530-nm (green) at the indicated light intensity of the 365-nm LED. Inactivation currents were fitted with an exponential equation giving τ_{inact} values at 365 nm. **b**, Superimposed traces recorded from another cell of the second and third light-gated currents time-locked to light activation (arrow head). **c**, Average τ_{inact} (left), time-to-peak (middle) and current-density (right) values ($n = 7$ cells). Indicated P -values are from paired t -test for time-to-peak and τ_{inact} data, and from Wilcoxon matched-pairs signed rank test for current amplitude data. **d**, Typical traces of whole-cell inward currents evoked by two cycles (I_1 and I_2) of irradiation at 365-nm (200 ms) and 530-nm light (800 ms) separated by different time intervals of 5, 10, 20, 40 and 60 s. Each trace was recorded from a different cell. **e**, Average ratio (I_2/I_1) of currents induced at 365 nm in function of time interval ($n = 11-23$ cells). Indicated P -values are from Wilcoxon test. **f**, Typical traces of whole-cell inward currents evoked by two cycles of irradiation at 365-nm (200 or 50 ms) and 530-nm light (800 or 950 ms) separated by a time interval of 10 s. Note the absence of current inactivation during the 50-ms irradiation at 365 nm. **g**, Average ratio (I_2/I_1) of currents in function of irradiation time at 365 nm ($n = 11-15$ cells). Mann-Whitney test, P -value *** = 0.0004. All data are presented as mean \pm s.e.m..

We also modified the text (line 119) as follows “However, we found that decreasing light intensity of the 365-nm LED significantly increased the time to reach the light-gated current peak (297 ± 20 ms), with a concomitant 2.3-fold decrease of current amplitude

(**Supplementary Fig. 4a-c**), suggesting that light intensity tunes both activation kinetics and current magnitude”, and in line 170 “Likewise, decreasing light intensity of the 365-nm LED significantly increased τ_{inact} values, suggesting that light-induced channel inactivation also depends on light intensity (**Supplementary Fig. 4a and c**)”.

We also added the following in the Discussion (line 383): “The second limitation of mOP1 is the relatively slow light-induced transitions as compared to faster mechanically-induced processes. One possibility is that different gating mechanisms are at play. However, this hypothesis seems unlikely as we found that changing light intensity tuned both light-induced activation and inactivation kinetics. Our data, therefore, support the idea that light-gated transitions can be accelerated to rates approaching those induced by mechanical stimulations by using more powerful irradiation devices”.

3) *Figure 1d shows that the inactivation of WT is similar to that of Y2464C, but in Extended Data Figure 4a, the fluorescence signal decrease of Y2464C under the Yoda1 is significantly faster than that of WT. How to explain this difference?*

We thank the reviewer for pointing out this issue. Because this point was also raised by reviewer #2, you will find a copy of the response below.

We thank the reviewer for pointing out this issue for which we take the opportunity to clarify it. First of all, we recognize that our initial manuscript was not sufficiently clear to understand that our calcium imaging assay does not contain washing steps. In these conditions, Yoda1 cannot be washed out, and thus this explains why mOP1 was constitutively potentiated by Yoda1. Second, it is true that the fluorescence signal decrease of Y2464C under Yoda1 was faster than that of wild-type mP1 (this point was also raised by reviewer #3, see point 3). This difference can actually be explained by a faster inactivation rate of the Y2464C mutant itself (not labelled by MAT) as compared to that of mP1 (see Supplementary Fig. 3f and g). If one assumes that labelling yield is low (which is likely to be the case here), then most of Yoda1-induced calcium responses in our system came from unlabeled channels. However, as shown in Supplementary Fig. 3f and g the difference in inactivation rates between the wild-type and Y2464C becomes very small, although significant, when both constructs were incubated with

MAT. Therefore, our data show that MAT labeling tends to normalize inactivation rates, which further supports the similarity of channel inactivation between mOP1 and mP1.

We have therefore modified the text in the revised manuscript as follows (line 233): “This plateau was expected because of the continuous bath presence of Yoda1 that constitutively gated PIEZO activity (there were no washing steps allowing Yoda1 removal for all experiments). In line with the fact that the MAT-unlabeled Y2464C mutant inactivated more rapidly than mP1 (**Supplementary Fig. 3f and g**), the decrease of fluorescent signal was also more rapid for mOP1 than for mP1. These data thus suggest that most of Yoda1-induced Ca²⁺ responses come from unlabeled channels”.

We hope this issue is resolved.

4) *Does MAT covalent binding affect the distribution of Piezo channels on the cell membrane?*

We thank the reviewer for this comment. Because this point was also raised by reviewer #2, you will find a copy of the response below.

This point was also raised by reviewer #3 (see point 4). We have thus performed additional experiments to investigate whether or not MAT labeling induces any changes of PIEZO1 membrane expression. By determining cell-surface expression of biotinylated Y2464C PIEZO1-HA labelled with MAT for 20 or 30 min, we clearly provide evidence that labeling at 30 min does not change plasma membrane expression, suggesting that long incubation times >20 min might induce a defect on channel gating rather than channel internalization.

Supplementary Fig. 7. Cell-surface protein biotinylation assay of Y2464C PIEZO1-HA. **a**, Uncropped Western blot of biotinylated membrane-associated proteins (cell-surface) or total cell lysate (input) separated by SDS-PAGE from HEK-P1KO cells transiently transfected with 5, 7.5 or 10 µg of plasmid encoding hemagglutinin (HA) tagged Y2464C mPIEZO1 (Y2464C PIEZO1-HA). The blot was probed with anti-HA antibody. Note the presence of specific bands running at the expected molecular masses (~300 kDa), as previously described^{2,3}. Below is shown the same blot re-probed with anti-β-Actin. **b**, Uncropped Western blot of biotinylated membrane-associated proteins (cell-surface) or total cell lysate (input) separated by SDS-PAGE from HEK-P1KO cells transiently transfected with 5 µg of plasmid encoding Y2464C PIEZO1-HA. Before biotinylation, transfected cells were incubated with 200 µM MAT for 20 or 30 min (n = 2 independent replicates). Blots were probed with anti-HA antibody. Below input is shown the same blot re-probed with anti-β-Actin. Apparent molecular weights are indicated on the left of Western blots. NT, non-transfected.

The mechanism by which this reduction occurs is currently unknown, but as we show that MAT is not active in *trans* (no light-induced currents), it is possible that such reduction might be a consequence of the chemical modification of the Y2464C side chain, rather than active *trans*-MAT. Regardless the mechanism, we provide evidence that no reduction of PIEZO1 activity occurred at 20 min incubation, which was the time incubation used throughout the manuscript.

We have now added these data in a new Supplementary Fig. 7 and modified the text as follows (line 274): “This reduction of activity was not related to a change of PIEZO1 plasma membrane expression, as assessed by cell-surface protein biotinylation (**Supplementary Fig. 7**), suggesting that long incubation times >20 min might induce a reduction of PIEZO1 gating through an unknown mechanism.”

5) MAT-treated cells expressing mPI also produce small light-gated responses in the presence of Yoda1 (Extended Data Figure 4). So how much of the increase in calcium response

of mOP1 after application of Yoda1 is due to the effect of this non-specific signal? it is recommended to add a control of mOP1 without MAT treatment.

As suggested by the reviewer, we have performed the requested control experiments (see also our response to comment 4 of reviewer #2). You will find below our response to comment 4 of reviewer #2.

First of all, it has to be stressed that these artifacts only occurred in the presence of Yoda1, not in its absence, as demonstrated by controls carried out in the absence of Yoda1 (mP1 treated with MAT) which revealed no signal in response to light (see Fig. 4c). However, as also suggested by reviewer #3 (see comment 5), **we have added new control experiments, in which light-gated calcium signals were monitored in Y2464C without MAT treatment both in the absence and presence of Yoda1 and with mOP1 using Yoda1 as reference (now shown in Supplementary Fig. 5a and c of the revised manuscript). Clearly no signal was detected in the absence of Yoda1 confirming the high specificity of mOP1.** In the presence of Yoda1, and as already described in the initial version, we observed these artifactual signals in the absence of MAT treatment (both in mP1 and Y2464C) and only in the presence of Yoda1. Therefore, these controls demonstrate that these artifactual signals were independent of MAT and only occurred in the presence of Yoda1. We also show that mOP1 signals in presence of Yoda1 were significantly higher than controls performed without MAT. How irradiation at 365 nm induced Yoda1-dependent calcium responses remains, however, unexplained.

Supplementary Fig. 5. GCaMP6 Ca^{2+} imaging with mOP1. **a**, Time-lapse of fluorescence variation (arbitrary units) depicting intracellular Ca^{2+} dynamic in response to 1 s irradiations at 365-nm in cells expressing Y2464C treated (thick black trace) or nor (dotted grey trace) with MAT. Note 1-s pre-irradiations at 530 nm just before the second and third irradiation at 365 nm. Yoda1 (10 μM) was applied as control (grey shadow). Corresponding fluorescent images are shown on the top at the indicated times. Scale bars, 10 μm . **b**, Time-lapse of fluorescence variation (arbitrary units) depicting intracellular Ca^{2+} dynamic in the continuous presence of 10 μM Yoda1 (arrow, applied just before the onset of the experiments) in response to 1 s irradiations at 365-nm in MAT-treated cells expressing either mOP1 (thick cyan trace) or mP1 (dotted cyan trace). Ionomycin (5 μM) was applied as control (grey shadow). Corresponding fluorescent images are shown on the top at the indicated times. Scale bars, 10 μm . **c**, Box plot of light-gated Ca^{2+} peak responses normalized to that of ionomycin or Yoda1 in function of irradiation cycles in the absence or presence of Yoda1. Responses are presented as median (center), 25-75 percentile (box) and 5-95 percentile (whisker). Outliers extending beyond whiskers (1.25 \times whisker length) are shown. Indicated P -values are from Mann-Whitney test. Number of cells is indicated in parentheses. Note the presence of unspecific, light-gated responses only in the presence of Yoda1 and in the absence of MAT (-MAT, untreated cells) expressing either Y2464C or mP1.

We have thus modified the text as follows (line 242): “Surprisingly, significant light-gated responses were also detected in MAT-treated cells expressing mP1 ($3.1 \pm 0.5\%$ of ionomycin signal, $n = 108$ cells); however, these signals appeared to be independent of MAT as they were observed in the absence of labeling in both Y2464C and mP1, and only in the presence of Yoda1 (Supplementary Fig. 5c)”.

We would like to keep the Yoda1 section in the revised form of the manuscript because it documents on the limits of the use of Yoda1 when irradiated at 365 nm. We hope these additional control experiments resolve this issue.

6) *In line 231, the conclusion that the $23.6 \pm 7.6\%$ channels is labelled cannot be simply concluded from the comparison of the maximal currents under the two different stimulation modalities.*

We agree with this comment, and accordingly, we have removed the sentence related to the labeling yield from the revised manuscript. Instead we used “light-gated current yield” (see line 272).

7) In the single-channel recording by irradiation, the conductance of o1, o2, and o3 are linearly proportional. Is it possible that o1, o2, and o3 correspond to 1, 2, and 3 channels, respectively, instead of the 3 states of one channel? How about the single channel conductance of Y2464C under mechanical stimulation? Furthermore, in figure 3e, it can be observed that mP1 also has flickering events that are smaller than full opening under pressure. Are these events different from o1 or o2 of mOP1?

If o1, o2 and o3 correspond respectively to 1, 2 and 3 channels, then o2 and o3 openings must correspond to the simultaneous opening of two and three channels, respectively. If that were the case, then one should expect to see currents with a “staircase”-like appearance, corresponding to the progressive opening of the channels. This is not what we observed, since o2 and o3 open in an all-or-nothing manner, with no evidence of staircase-like openings (see **Figure R1** below for relevant examples for the o3 state). Our data thus rather support our initial hypothesis that o1, o2 and o3 states belong to one channel.

Figure R1. Typical examples of single-channel mOP1 currents at -80 mV in the cell-attached configuration elicited by 365 nm. Traces are sections of recordings shown in Fig. 5f and Supplementary Fig. 6e of the revised

manuscript at high time resolution showing single events of putative o3 state (with actual values above traces). Label c denotes the closed conductance level (data are from four different cell-attached patches). Note the absence of staircase-like openings.

Regarding flickering events in mP1, it is possible that wild-type channels also display distinct conductance states, but their contributions seem to be low. We believe that such analysis would need additional experiments that would go beyond the scope of the present study.

Finally, as suggested by the reviewer, we **have carried out additional experiments to show that single-channel conductance of Y2464C under mechanical stimulation was similar to that of mP1 and to that of o3 in mOP1**. We have added these new data in Figure 5g and a relevant trace in Supplementary Fig. 6h. Consistent with the fact that Y2464C inactivated more rapidly than the wild-type mP1, brief single-channel openings were recorded, but this was not analyzed further.

Figure 5. Single-channel currents of mOP1. **a**, Typical inward currents at -80 mV in the cell-attached configuration evoked by first applying negative pressure through the recording pipette (red arrow in inset) and then light irradiation 10 s after. Shown are two patches from MAT-treated cells expressing either mOP1 (upper traces) or mP1 (bottom traces). A pre-pulse of +5 mm Hg was applied to the patch to minimize resting membrane tension, as previously described³². **b**, Average light-gated (violet histograms) and pressure-evoked (black histograms) currents on mOP1 and mP1 (n = 14 cells for mOP1 and 8 for mP1). **c**, Average time-to-peak activation data. **d**, Current dependency as a function of incubation time of cells with 200 μ M MAT (n = 8, 16, 14 and 15 cells for incubation times 0, 10, 20 and 30 min, respectively). Comparison with Mann-Whitney test. *P*-values: * = 0.0283, # = 0.0011, ** = 0.0047, *** = 0.0001. **e**, Typical single-channel currents at -80 mV in the cell-attached configuration elicited by light irradiation or pressure stimulation of cells expressing either mOP1 (MAT treated) or mP1 (not treated). Channel openings are downward deflections. **f**, Sections of the recordings shown in **e** (cyan lines below traces) are displayed at higher time resolutions. Labels c and o denote, respectively, closed and open conductance levels of mP1 and o₁, o₂ and o₃ represent the three open conductance levels of mOP1. The violet arrow indicates the start of irradiation. Corresponding all-point histograms with Gaussian fits are shown right of the traces. Data were filtered at 1 kHz. **g**, Average single-channel conductance determined at -80 mV (n = 15 cell-attached patches for mOP1, 6 for Y2464C (MAT not treated), and 9 for mP1) for light (violet data) and pressure (black data) stimulations. Comparison with unpaired *t*-test. *P*-values: *** < 0.0001. NS, not significant (*P* = 0.4817).

between o_3 and o mP1). Comparison with Mann-Whitney test for Y2464C data. $P = 0.5622$ between o_3 and o Y2464C, and $P = 0.5287$ between o Y2464C and o mP1). All data are presented as mean \pm s.e.m..

Supplementary Fig. 6. Functional characterization of pressure- and light-evoked responses in the cell-attached configuration. **a**, Typical inward currents in the cell-attached configuration evoked by applying negative pressure through the recording pipette (-60 mm Hg for 300 ms) at a holding potential of -80 mV. Shown are two patches from cells that were transfected (upper trace) or not (lower trace) with mP1. To minimize resting membrane tension, a pre-pulse of +5 mm Hg was applied to the patch for 5 s prior to mechanical stimulations, as previously described¹. **b**, Current-pressure relationship of stretch-activated currents at -80 mV from patches expressing the indicated construct treated or not with MAT. Data were fitted with a Boltzmann equation giving I_{max} and P_{50} ($n = 11-13$ patches, indicated in parentheses). **c**, **d**, Corresponding I_{max} and P_{50} values. P -values are from Mann-Whitney test. **e**, Left, examples of single-channel currents at -80mV in the cell-attached configuration elicited by light irradiation (365 nm and 530 nm) from MAT-treated cells expressing mOP1. Channel openings are downward deflections. Right, sections of the recordings indicated by cyan lines below traces are shown at a higher time resolution. Label c denotes closed channels and labels o_1 , o_2 and o_3 represent the three open conductance levels. Violet arrows indicate the start of irradiations. Corresponding all-point histograms with Gaussian fits are shown right of the traces. Data were acquired at 40 kHz and filtered at 1 kHz. **f**, Light-induced conductance (at -80 mV) of o_1 , o_2 and o_3 as a function of labeled subunits ($n = 15$ cell-attached patches). Each cell-attached patch is shown by a different symbol and color. Data were fitted with a linear regression (violet line) of the form $y = 7.5 \times x - 1.1$. Inset: cartoon depicting subunit labeling stoichiometry with one, two and three MAT molecules putatively corresponding, respectively, to o_1 , o_2 and o_3 . All data are presented as mean \pm s.e.m.. **g**, Distribution of detected events obtained from all cell-attached patches ($n = 15$) as a function of current amplitude.

Amplitude histogram was fitted with three Gaussians and fitted values are shown on the top. Note the presence of three conductance level values that were very similar to those determined individually from each patch. **h**, Single-channel currents at -80 mV in the cell-attached configuration elicited by pressure from MAT-untreated cells expressing Y2464C. Channel openings are downward deflections. Labels c and o denote, respectively, closed and open conductance levels. Corresponding all-point histogram with Gaussian fits is shown right of the trace. Data were acquired at 20 kHz and filtered at 1 kHz.

We have modified the text as follows (line 293): “O₁ and o₂ were significantly different from o₃, while o₃ was not different from the conductance level (o) of unlabeled mP1 or Y2464C stimulated with pressure (**Figure 5g and Supplementary Fig. 6h**), suggesting that o₃ would correspond to full channel opening with a single channel conductance that was not far from those previously reported^{2,7}”.

8) *Given that MAT can bind cysteine, does MAT affect the crosslinking of cysteine between the cap and the blade? Does DTT affect the binding of MAT to Y2464C? Besides illumination, it is better to show the currents under the mechanical stimulation in Figure 4b.*

We thank the reviewer for raising this issue, for which we have carried out additional experiments to show that the triple mutant R1762C/E2257C/Y2464C is functional to poking stimulations in the presence of DTT, both in the absence or presence of MAT. As previously found for Y2464C (Supplementary Fig. 3g), we show that inactivation kinetics were significantly faster in MAT-untreated R1762C/E2257C/Y2464C than in MAT-treated R1762C/E2257C/Y2464C. These data therefore suggest that DTT does not affect MAT labeling to Y2464C (as expected for covalent bonds formed between maleimides and cysteines), and that MAT labeling does not prevent mechanical-induced responses. Data are now shown in Supplementary Fig. 8.

Supplementary Fig. 8. Poking-evoked currents in R1762C/E2257C/Y2464C in the presence of DTT. a, Typical whole-cell currents recorded at -80 mV elicited by cell poking in cells treated or not by MAT. DTT (10 mM) was bath applied at least 3 minutes before recordings. Inactivated currents were fitted with an exponential equation giving τ values. **b**, Average τ , time-to-peak and current density values in the absence (grey) or presence

(black) of **MAT** (n = 6-10 cells). Unpaired t test (for τ and time-to-peak) or Mann-Whitney test (current density). P-values: *= 0.0314, NS = 0.4663 and 0.5622 (from left to right). NS, not significant.

We have thus modified the text as follows (line 315): “and robust poking-evoked currents were recorded in the presence of DTT in R1762C/E2257C/Y2464C with inactivation kinetics that depended on **MAT** labeling (**Supplementary Fig. 8**), as observed for the Y2464C mutation (**Supplementary Fig. 3g**)”.

Regarding the comment suggesting that we should show mechanical-induced currents besides illumination, I am afraid that we cannot satisfy the request of the reviewer, because we did not record paired light-gated currents with poking-evoked currents. The reason for this is that these experiments are extremely challenging to carry out (i.e. alternating cell-poking and irradiation stimulations in patched cells in which the extracellular solution was alternatively exchanged between DTT-containing and DTT free buffers). We hope the reviewer understands.

9) In line 302, Gd^{3+} is generally thought to affect MS channels by affecting cell membrane property, so it may be more reasonable to use a more recognized pore blocker (e.g. ruthenium red) for the Δ blade mutants.

As suggested by the reviewer, we have added ruthenium red (RR) experiments and found that RR inhibited leaky currents in cells expressing Δ blade mP1 and Δ blade mOP1, but not basal currents obtained in non-transfected cells.

Supplementary Fig. 11. Specific reduction of holding currents upon Gd^{3+} or RR application in cells expressing truncated PIEZO1 channels. **a.** Representative whole-cell currents at -80 mV of Δ blade mOP1 (left) and Δ blade mP1 (right) in response to light irradiation and 30 μ M RR application from cells treated with **MAT**. **b.** Representative whole-cell currents from non-transfected cells recorded during a 30- μ M Gd^{3+} (top) or 30- μ M

RR (bottom) application. Note the absence of leaky currents. **c**, Average Gd^{3+} - and RR-induced inhibitions of holding currents recorded in cells transfected with Δ blade mP1 or Δ blade mOP1 (mean \pm s.e.m., n = 7-9 cells). NT, non-transfected cells, RR, ruthenium red.

We have now added these data in Supplementary Fig. 11 and revised the manuscript as follows (line 344): “To detect spontaneous pore activity, we used gadolinium (Gd^{3+}) and ruthenium red (RR), two known mP1 pore blockers¹, and found specific reduction of the holding current upon blocker application that was not observed in non-transfected cells (**Figure 6e and Supplementary Fig. 11**)”.

10) *The significant difference of inactivation time between light-gated current and mechanical-gated current suggests different gating mechanisms. What makes this difference?*

We provide evidence that light intensity makes this difference (see our response of point 2 of reviewer #3). Because light intensity of our LED was set to its maximum, it remains to be tested whether more powerful devices (such as lasers, for which our lab is not equipped) can increase whole-cell light-gated current amplitudes and decrease τ_{inact} .

11) *The data of the Δ blade mutants can only prove that the blade can promote the mechanical transduction, but cannot prove that the blade can directly sense photoisomerization, and there may be a regulatory effect. Thus, the authors should compare the response with and without blade, and the subtitle in line 278 is less accurate.*

We thank the reviewer for his/her helpful comment. We agree that our initial wordings were not accurate enough. We have now revised the text in the results and discussion.

The subtitle was changed to (line 322): “**The blades contribute to the rapidity of light-induced channel opening**”

12) *The blades sense membrane perturbations caused by azobenzene photoisomerization to trigger channel opening. This seems to overestimate the necessity of blades in the light-gated motions, since Δ blade mOP1 is still able to respond to 365-nm light and it is unclear whether MAT interacts with phospholipids. Is it possible that the blades, as part of the Piezo channel,*

just assist in the opening of the central pore rather than directly sensing azobenzene photoisomerization ?

We agree with this comment. We have thus changed Fig 6i by substituting “1. Local membrane perturbation. 2. Rapid blade-sensing” by a question mark to highlight a possible hypothesis.

We have also modified the Discussion as follows (line 390): “We provide evidence that the blades are not necessary for light-gated motions, but they are critically involved in the rapidity of light-induced channel opening. It is unclear how the blades accelerate light-induced channel opening, but it might be speculated that *trans*-to-*cis* azobenzene photoisomerization can induce a local membrane tension perturbation that is rapidly sensed by the blades, which, in turn, respond by triggering channel opening (**Figure 6i**)”.

13) *“To detect spontaneous pore activity, we used Gd3+, a known mP1 pore blocker1, and found specific reduction of the holding current upon Gd3+ application that was not observed in non-transfected cells (Figure 4e and Extended Data Figure 6).” The image and text do not match.*

We thank the reviewer for pointing this out. We have corrected it.

Other minor comment:

In line 304, the figure should be Extended Data Figure 8 instead of Extended Data Figure 6.

We thank the reviewer for pointing this out. We have corrected it.

Reviewers' Comments:

Reviewer #1:

Remarks to the Author:

The revised manuscript has been strengthened and the reviewers have addressed our concerns.

Reviewer #3:

Remarks to the Author:

The authors have addressed the comments and the reviewer support the publication of the manuscript.

A point-by-point response to the reviewers' comments is shown below. Our response is shown in blue.

Reviewer #1 (Remarks to the Author):

The revised manuscript has been strengthened and the reviewers have addressed our concerns.

We thank the reviewer for his/her positive and enthusiastic feedback.

Reviewer #3 (Remarks to the Author):

The authors have addressed the comments and the reviewer support the publication of the manuscript.

We thank the reviewer for his/her positive comment and for supporting the publication of our work.